# On the Performance Analysis of Momentum Method: A Frequency Domain Perspective

**Xianliang Li**[*1,2], **Jun Luo**[*1,2], **Zhiwei Zheng**[*3], **Hanxiao Wang**[2,4],
**Li Luo**[5], **Lingkun Wen**[2,6], **Linlong Wu**[7], **Sheng Xu**[†1]

[1]Shenzhen Institutes of Advanced Technology, Chinese Academy of Sciences
[2]University of Chinese Academy of Sciences  [3]University of California, Berkeley
[4]Institute of Automation, Chinese Academy of Sciences  [5]Sun Yat-sen University
[6]Shanghai Astronomical Observatory, Chinese Academy of Sciences  [7]University of Luxembourg
`yinleung.ley@gmail.com`, `{j.luo3,sheng.xu}@siat.ac.cn`,
`zhiwei.zheng@berkeley.edu`, `wanghanxiao18@mails.ucas.ac.cn`,
`luoli33@mail2.sysu.edu.cn`, `wenlingkun@shao.ac.cn`, `linlong.wu@uni.lu`

## Abstract

Momentum-based optimizers are widely adopted for training neural networks. However, the optimal selection of momentum coefficients remains elusive. This uncertainty impedes a clear understanding of the role of momentum in stochastic gradient methods. In this paper, we present a frequency domain analysis framework that interprets the momentum method as a time-variant filter for gradients, where adjustments to momentum coefficients modify the filter characteristics. Our experiments support this perspective and provide a deeper understanding of the mechanism involved. Moreover, our analysis reveals the following significant findings: high-frequency gradient components are undesired in the late stages of training; preserving the original gradient in the early stages, and gradually amplifying low-frequency gradient components during training both enhance performance. Based on these insights, we propose *Frequency Stochastic Gradient Descent with Momentum* (FSGDM), a heuristic optimizer that dynamically adjusts the momentum filtering characteristic with an empirically effective dynamic magnitude response. Experimental results demonstrate the superiority of FSGDM over conventional momentum optimizers. [1]

## 1 Introduction

Momentum has achieved great success in deep learning applications when combined with *Stochastic Gradient Descent* (SGD) (Robbins & Monro, 1951). Among various momentum methods (Polyak, 1964; Nesterov, 1983; Van Scoy et al., 2017; Ma & Yarats, 2018; Kidambi et al., 2018), one of the most prevalent variants is the momentum method utilized within *Stochastic Gradient Descent with Momentum* (SGDM) (Sutskever et al., 2013; Paszke et al., 2019), which can be expressed as:

$$\text{Standard-SGDM (decoupled)}: m_t = u_t m_{t-1} + v_t g_t, \quad x_t = x_{t-1} - \alpha_t m_t, \tag{1}$$

where $g_t$ denotes the gradient at iteration $t$, $m_t$ is the momentum buffer, and $x_t$ represents the learnable parameters. The momentum coefficients $u_t$ and $v_t$ control the influence of the previous momentum and the current gradient, respectively, and $\alpha_t$ is the learning rate. For these time-variant momentum coefficients, a multistage setting has been commonly adopted in the machine learning community (Aybat et al., 2019; Kulunchakov & Mairal, 2019; Liu et al., 2020). Throughout this paper, we refer to this formulation, which decouples the two momentum coefficients, as Standard-SGDM. In contrast, another prevalent variant couples the two momentum coefficients using the *Exponential Moving Average* (EMA) method (Gardner Jr, 1985), leading to the formulation of EMA-SGDM:

$$\text{EMA-SGDM (coupled)}: m_t = u_t m_{t-1} + (1 - u_t) g_t, \quad x_t = x_{t-1} - \alpha_t m_t, \tag{2}$$

---

∗: Equal contribution. †: Corresponding author.
[1]Our implementation of FSGDM is available at `https://github.com/yinleung/FSGDM`.

where $u_t \in [0, 1)$ is the momentum coefficient. Notably, this coupled momentum formulation is a special case of the decoupled one, i.e., Standard-SGDM with $v_t = 1 - u_t$. Our experiments show performance gaps between these two formulations. Moreover, how the momentum coefficients change over time can significantly affect the test accuracy (see Section 3). The existence of these two distinct momentum formulations and their differing performances raise two primary questions in modern deep learning:

1. **Decoupling vs. Coupling**: Should the coefficients $u_t$ and $v_t$ be decoupled or coupled?

2. **Temporal Variation**: How should the momentum coefficients evolve over time during training to achieve better model performance?

For Question 1, some literature has investigated the convergence of the coupled method (Mai & Johansson, 2020; Li et al., 2022). Liu et al. (2020) argued that coupling the coefficients leads only to a constant scaling difference. Wang et al. (2024) further demonstrated that the mathematical equivalence between EMA-SGDM and Standard-SGDM can be achieved by adjusting the momentum coefficients and the learning rates in a coupled way. However, in practice, learning rate schedules are typically independent of momentum coefficient tuning during network training. On the other hand, popular frameworks like PyTorch (Paszke et al., 2019) adopt a decoupled momentum strategy by default. In our framework, we tackle the first question from the frequency domain perspective, revealing the relationship between the coupled and decoupled constructions.

Regarding Question 2, prior research offered diverse opinions on how the momentum coefficients should vary over time. Some studies preferred fixed decoupled momentum coefficients (Yan et al., 2018; Liu et al., 2018; Yu et al., 2019), commonly selecting $u_t$ values as 0.9 and $v_t$ value as 1. Liu et al. (2020) highlighted the benefits of stagewise learning rate schedules in EMA-SGDM, noting that $u_t$ can either remain constant or increase along with the stagewise adjustments. Conversely, Smith (2018) demonstrated that decreasing the momentum coefficients while increasing the learning rate improves test performance. Moreover, Adaptive momentum methods (Kingma & Ba, 2014; Reddi et al., 2018; Luo et al., 2019; Chen et al., 2018) proved the convergence of decreasing coupled momentum coefficients in the context of online convex optimization. Nonetheless, a consensus regarding the optimal time-variant pattern of the momentum coefficients has yet to be reached.

To answer these questions, one has to understand how the momentum method affects the training process. Goh (2017) analyzed the momentum method from the aspect of convergence and dynamics. Several prior studies (Cutkosky & Orabona, 2019; Ma & Yarats, 2018) speculated that averaging past stochastic gradients through momentum might reduce the variance of the noise in the parameter update, thus making the loss decrease faster. Polyak (1964); Rumelhart et al. (1986) argued that the EMA momentum can cancel out oscillations along high-curvature directions and add up contributions along low-curvature directions. From the signal processing perspective, the EMA method acts as a discrete low-pass filter for smoothing out high-frequency fluctuations while retaining the low-frequency baseband pattern of the signal (Gardner Jr, 1985). These points of view bring us a new insight into connecting the momentum update processes with the specific filters. In this aspect, the momentum methods with different coefficient selections can be interpreted in a unified frequency domain analysis framework, whereby Questions 1 and 2 are resolved.

In this paper, we propose a novel frequency domain analysis framework to address the two questions and provide a deeper understanding of the role of momentum in stochastic optimization. To the best of our knowledge, this paper, for the first time, reveals the fundamental difference between Standard-SGDM and EMA-SGDM and uncovers the effects of the dynamic momentum coefficients clearly from the frequency domain perspective. This perspective not only explains the difference between various momentum methods but also provides practical guidelines for designing efficient optimizers. Accordingly, we introduce FSGDM, an optimizer that dynamically adjusts momentum filter characteristics during training. Experiments show that FSGDM outperforms traditional SGD-based momentum optimizers.

## 2 FREQUENCY DOMAIN ANALYSIS FRAMEWORK

This section introduces the background of Z-transform (Zadeh, 1950) in signal processing and then proposes a new frequency domain analysis framework for momentum methods.

## 2.1 Z-Transform and Quasi-Stationary Approximation

Frequency analysis is a crucial technique for understanding how systems react to varying frequency components of input signals. Specifically, for discrete-time linear time-invariant systems, Z-transform is leveraged to examine how systems attenuate or amplify signals at specific frequencies, especially in the study of system stability, pole-zero behavior, etc. (Oppenheim et al., 1996).

Interestingly, in neural network training, the momentum update process at time $t$ can be seen as a recursive filter where the gradient $g_t$ and the momentum $m_t$ act as input and output signals, respectively. The momentum coefficients affect the gradient adjustments across different frequency components. The high-frequency gradient components correspond to large and more abrupt changes in the gradient; while the low-frequency components indicate smooth and more gradual adjustments.

However, one key issue is that the momentum system can be inherently time-variant, as its coefficients may change stagewise throughout the training process. This variability makes it difficult to apply traditional Z-transform analysis. To overcome this, inspired by the Zadeh (1961); Jury (1964), we approximate the system as time-invariant in each discrete interval stage. By holding the momentum coefficients constant over every interval, we construct a time-invariant quasi-stationary system (Hubner & Tran-Gia, 1991), enabling us to apply the Z-transform validly.

In our following analysis framework and our later optimizer design, we follow this multistage strategy for changing momentum coefficients. Particularly, for a predefined stage whose length is denoted by $\delta$, the momentum coefficients are redefined using the floor function to ensure they remain constant over the whole stage:

$$u_t = u(\lfloor t/\delta \rfloor \times \delta) \quad \text{and} \quad v_t = v(\lfloor t/\delta \rfloor \times \delta), \tag{3}$$

where $u(t), v(t)$ are the continuous dynamic sequence functions with respect to $t$. While there are multiple sequences with different designs, in this paper, we use the following increasing and decreasing sequences:

$$\text{Increasing}: u(t) \text{ or } v(t) = \frac{t}{t+\mu}, \quad \text{Decreasing}: u(t) \text{ or } v(t) = 1 - \frac{t+1}{t+\nu}, \tag{4}$$

where $\mu$ and $\nu$ are the increasing and decreasing factors [2]. In Appendix C.1, we also examined the test set performance using other kinds of dynamic sequences. Under the above settings, for a given stage $k$ $(k = 1, \cdots, N)$, with $t \in [(k-1)\delta, k\delta - 1]$, the momentum system becomes:

$$m_t = u_k m_{t-1} + v_k g_t \tag{5}$$

where $u_k = u((k-1)\delta)$ and $v_k = v((k-1)\delta)$ are constants for the duration of the $k$-th stage. Additionally, we set the total number of stages, denoted by $N$, to a constant value of 300 for all the experiments in this paper.

## 2.2 Frequency Domain Analysis of the Momentum Method

In this subsection, we introduce our frequency domain analysis framework and analyze the impacts of the momentum method on neural network training. We first apply Z-transform, denoted by $\mathcal{Z}$, to Equation 5:

$$M(z) = u_k z^{-1} M(z) + v_k G(z), \tag{6}$$

where $G(z) = \mathcal{Z}\{g_t\}$, $M(z) = \mathcal{Z}\{m_t\}$, and $z^{-1}M(z) = \mathcal{Z}\{m_{t-1}\}$. To obtain the frequency response of the momentum system during stage $k$, we evaluate the transfer function $H_k(z)$ on the unit circle (Oppenheim et al., 1996):

$$H_k(z) = \frac{M(z)}{G(z)} = \frac{v_k}{1 - u_k z^{-1}} \quad \xrightarrow{z = e^{j\omega}} \quad H_k(\omega) = \frac{v_k}{1 - u_k e^{-j\omega}}, \tag{7}$$

where $\omega \in [0, \pi]$ is the normalized angular frequency of the real-value signal. The frequency response of the momentum system describes how the input gradient signal $G(z)$ is altered to the output momentum signal $M(\omega)$ when it passes through the system. Note that this transfer function is valid for the entire duration of the $k$-th quasi-stationary stage.

---

[2]Note that different from the increasing sequence, the numerator of the decreasing sequence is $t + 1$. This design avoids the zero gradients at the first training stage.

**Magnitude Response.** The magnitude response of the momentum system in the $k$-th stage can be calculated by taking the magnitude of $H_k(\omega)$:

$$|H_k(\omega)| = \frac{|v_k|}{\sqrt{1 - 2u_k \cos \omega + u_k^2}}. \tag{8}$$

The magnitude response describes the amplitude scaling effect of the system at different frequencies. It indicates how the momentum system amplifies or attenuates different frequency components during each stage. This characteristic of the momentum system plays a key role in affecting the optimization process. Notably, when $|H_k(\omega)| < 1$, the momentum system attenuates the signals with frequency $\omega$; when $|H_k(\omega)| > 1$, the momentum system amplifies the signals with $\omega$. Consequently, we divide the momentum systems into two categories: *Orthodox Momentum Systems* and *Unorthodox Momentum Systems*.

*Orthodox Momentum Systems* are the ones whose amplitude of the magnitude response will not surpass 1, like the EMA-SGDM (2). This kind of momentum system only shows attenuating characteristics. Specifically, the momentum system behaves as a **low-pass** filter when $u_k > 0$ and a **high-pass** filter when $u_k < 0$. Additionally, when $u_k$ gets close to 1, the momentum system will prefer to attenuate the gradient components with high frequencies. The visualization of the (dynamic) magnitude responses of orthodox momentum systems is in Section 3.1 and Appendix C.2.

For *Unorthodox Momentum Systems* where the amplitude of magnitude response will surpass 1, such as selecting $u_t = 0.9$ and $v_t = 1$ in Standard-SGDM (1), the momentum system possesses both amplifying and attenuating characteristics. In this paper, we refer to these kinds of unorthodox filters as **low/high-pass gain** filters. Specifically, the momentum system behaves as a low-pass gain filter when $u_k > 0, v_k = 1$ and a high-pass gain filter when $u_k < 0, v_k = 1$. Additionally, if $u_k$ is close to 1, the momentum system attenuates high-frequency gradient components while strongly amplifying low-frequency components; if $u_k$ is close to $-1$, the momentum system attenuates low-frequency gradient components while strongly amplifying high-frequency components. The visualization of the (dynamic) magnitude responses of unorthodox momentum systems is in Section 3.2 and Appendix C.2.

To demonstrate the momentum effects from the frequency perspective, in Figure 1, we compare an original sinusoidal signal, a noisy version injected with Gaussian noise, and the signal after applying the momentum method (which is called momentum signal for short) in the time domain. The red curve represents the noisy signal, the black dashed curve corresponds to the original noise-free true signal, and the cyan curve shows the momentum signal. We can see that different selections of $u_k$ and $v_k$ significantly affect the amplifying or attenuating effects of the momentum system.

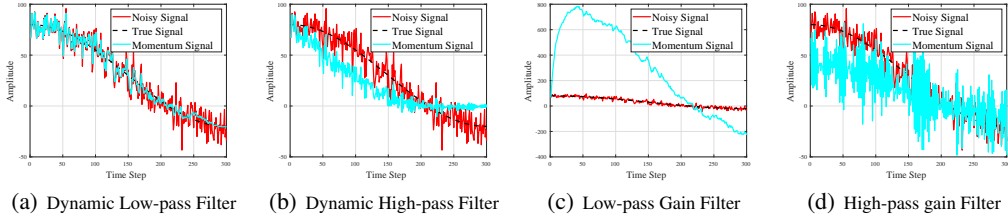

(a) Dynamic Low-pass Filter    (b) Dynamic High-pass Filter    (c) Low-pass Gain Filter    (d) High-pass gain Filter

Figure 1: Visualization of different filters towards the noisy sinusoidal signal. (a) $u_k = 0 \to 1, v_k = 1 - u_k$, with the system gradually shifting from an all-pass filter to a narrow low-pass filter; (b) $u_k = 0 \to -1, v_k = 1 + u_k$, with the system gradually shifting from an all-pass filter to a narrow high-pass filter; (c) $u_k = 0.9, v_k = 1$, which indicates the momentum behaves like a low-pass gain filter with amplification on low-frequency gradient components; (d) $u_k = -0.9, v_k = 1$, which indicates the momentum behaves like a high-pass gain filter with amplification on high-frequency components. The amplifying and attenuating effects of different momentum systems are verified.

Similarly, we also have the phase response of the momentum system (see Appendix A). While the phase response of the momentum only provides limited insights, understanding the behavior of the magnitude response across stages is essential for analyzing the time-variant characteristics of the momentum system. By plotting the dynamic magnitude response value $|H_k(\omega)|$ on the normalized

angular frequency axis for each stage $k$, we can track how the frequency-dependent behavior of the multistage momentum system evolves. This provides valuable insights into the amplifying or attenuating characteristics of the momentum system. Further results on the comparisons of momentum systems with different dynamic magnitude responses are presented in the next section.

## 3 DYNAMIC MAGNITUDE RESPONSE OF THE MOMENTUM SYSTEMS

In this section, we present an empirical study to discover the influence of the momentum coefficients by comparing the test performance on momentum systems with different dynamic magnitude responses. We train VGG (Simonyan & Zisserman, 2014) on the CIFAR-10 (Krizhevsky et al., 2009) dataset and ResNet50 (He et al., 2016) on the CIFAR-100 dataset using different momentum coefficients, while keeping all other hyperparameters unchanged. For each experiment, we report the mean and standard error (as subscripts) of test accuracy for 3 runs with random seeds from 0-2. The detailed experimental settings can be found in Appendix D. The experimental results in CIFAR-10 show high similarity to those in CIFAR-100. Thus, here, we mainly focus on the analysis based on CIFAR-100 and defer the experimental results of VGG16 on CIFAR-10 in Appendix C.3.

### 3.1 ORTHODOX MOMENTUM SYSTEMS

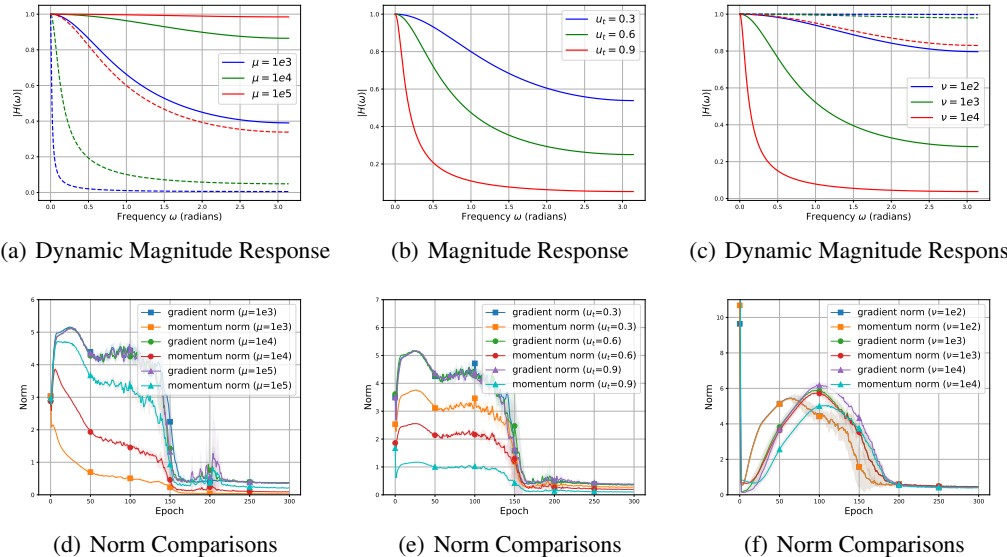

(a) Dynamic Magnitude Response  (b) Magnitude Response  (c) Dynamic Magnitude Response

(d) Norm Comparisons  (e) Norm Comparisons  (f) Norm Comparisons

Figure 2: (**Up**) Analysis of the (dynamic) magnitude responses in the early and late training stages for EMA-SGDM with low-pass momentum defined in Equation 9. The *solid lines* denote the magnitude responses in the *early stages*, and the *dashed lines* denote the magnitude responses in the *late stages*. (**Down**) The comparison between the gradient norms and momentum norms for EMA-SGDM with low-pass momentum. Left Column: increasing sequence. Middle Column: fixed sequence. Right Column: decreasing sequence.

We first focus on the orthodox momentum systems with the following two main types: low-pass and high-pass momentum, defined as:

$$\textbf{Low-pass}: m_t = u_t m_{t-1} + (1 - u_t)g_t, \quad \textbf{High-pass}: m_t = -u_t m_{t-1} + (1 - u_t)g_t, \quad (9)$$

where $u_t \in [0, 1)$ can be set as increasing, decreasing sequences, or fixed value. For time-variant momentum systems, different strategies of $u_t$ result in different time-variant filtering characteristics during training. According to Section 2.1, scaling the increasing and decreasing factors affects the changing rates of $u_t$. In the following, we demonstrate the dynamic magnitude responses, comparisons between gradient norms and momentum norms, and test accuracy results of orthodox momentum systems under different $u_t$ sequences [3].

---

[3]Note that selecting $\mu = 100$ and $\nu = 10^4$ lead to a long stage of the super narrow-band filter. To avoid this problem, we select $\mu = 10^3, 10^4, 10^5$ and $\nu = 10^2, 10^3, 10^4$ in this paper.

**Example 1: Low-Pass Momentum.** We first explore the effect of increasing, fixed, and decreasing $u_t$ sequences in low-pass momentum. Figure 2(a) - 2(c) show the corresponding dynamic magnitude responses over time. With increasing $u_t$, the system transits from an all-pass to a progressively narrower low-pass filter, gradually attenuating high-frequency components. Larger $\mu$ results in slower transitions. Decreasing $u_t$ shows a reverse behavior, with larger $\nu$ resulting in slower transitions. $u_t$ with a fixed value maintains a constant filter, with larger $u_t$ leading to more aggressive smoothing and noise reduction characteristics. The norm comparisons in Figure 2(d) - 2(f) show that the momentum norms in low-pass momentum systems are always less than corresponding gradient norms. Larger $u_t, \nu$ and smaller $\mu$ lead to more reduced momentum norms, which validates the time-variant filtering characteristics of orthodox momentum systems.

Test accuracy results in Table 1 reveal that increasing or fixing $u_t$ can achieve higher accuracy compared to applying decreasing sequences of $u_t$. In particular, momentum systems with proper increasing sequences of $u_t$ can outperform those with fixed $u_t$. We also find that larger $\nu$ results in poorer model performance. These phenomena indicate that gradually attenuating high-frequency components during training improves test set performance, while excessive suppression of low-frequency gradient components in early stages and retention of high-frequency components in late stages degrade model performance.

**Example 2: High-Pass Momentum.** High-pass momentum systems exhibit symmetric dynamic magnitude responses and similar norm comparisons, compared to their low-pass counterparts (see Figure 6 in Appendix C.2). With increasing $u_t$, the system shifts from an all-pass to a narrow high-pass filter, progressively attenuating low-frequency components. Decreasing sequences act in reverse. Fixed sequences with larger $u_t$ lead to more aggressive attenuation of low-frequency components. The comparison of gradient norms and momentum norms can be found in Appendix C.2.

Test accuracy in Table 1 shows that dynamic high-pass systems with larger $\mu$ and smaller $\nu$ yield better top-1 accuracy performance. When selecting fixed values, momentum systems with larger $u_t$ perform more poorly. These results confirm that suppressing low-frequency gradient components is harmful. Moreover, high-pass systems generally outperform low-pass systems when applying decreasing strategies with the same $\nu$, suggesting that high-frequency components play a crucial role in the early training stages, which is also supported by the studies in Appendix C.4.

From Examples 1 and 2, we empirically verify that high-frequency gradient components are detrimental in late training stages, while their preservation in early stages leads to higher test accuracy, which matches the viewpoint that gradient noise has a generalization benefit early in training (Smith et al., 2020).

Table 1: Top-1 ACC. (%) comparisons of different momentum coefficient strategies of orthodox momentum systems of ResNet50 on CIFAR-100.

| Parameters | Increasing Factor ($\mu$) | | | Fixed Value ($u_t$) | | | Decreasing Factor ($\nu$) | | |
|---|---|---|---|---|---|---|---|---|---|
| | 1k | 10k | 100k | 0.3 | 0.6 | 0.9 | 100 | 1k | 10k |
| Low-pass | $77.12_{0.07}$ | $77.06_{0.14}$ | $76.86_{0.12}$ | $76.98_{0.09}$ | $76.82_{0.18}$ | $76.84_{0.06}$ | $72.58_{0.44}$ | $70.53_{0.31}$ | $69.69_{0.75}$ |
| High-pass | $51.59_{0.78}$ | $67.55_{0.22}$ | $74.72_{0.06}$ | $72.46_{0.13}$ | $65.14_{0.17}$ | $53.43_{0.26}$ | $76.82_{0.25}$ | $75.92_{0.12}$ | $70.99_{0.18}$ |

## 3.2 Unorthodox Momentum Systems

Unorthodox momentum systems allow magnitude responses larger than 1, meaning they can both attenuate and amplify gradients in different frequency bands. We focus on two main types: low-pass gain and high-pass gain momentum, defined as:

$$\text{Low-pass gain}: m_t = u_t m_{t-1} + g_t, \quad \text{High-pass gain}: m_t = -u_t m_{t-1} + g_t, \quad (10)$$

where $u_t \in [0, 1)$ can follow increasing, fixed, or decreasing sequences. For simplification reasons, we use the PyTorch setting with $v_t = 1$. We show the dynamic magnitude responses, comparisons between gradient norms and momentum norms, and test accuracy results of unorthodox momentum systems under different $u_t$ sequences as follows.

**Example 3: Low-Pass Gain Momentum.** In low-pass gain momentum, the system transits from an all-pass to a narrower low-pass gain filter as $u_t$ increases, amplifying low-frequency components while attenuating high-frequency components. Figure 3(a) - 3(c) show the corresponding dynamic

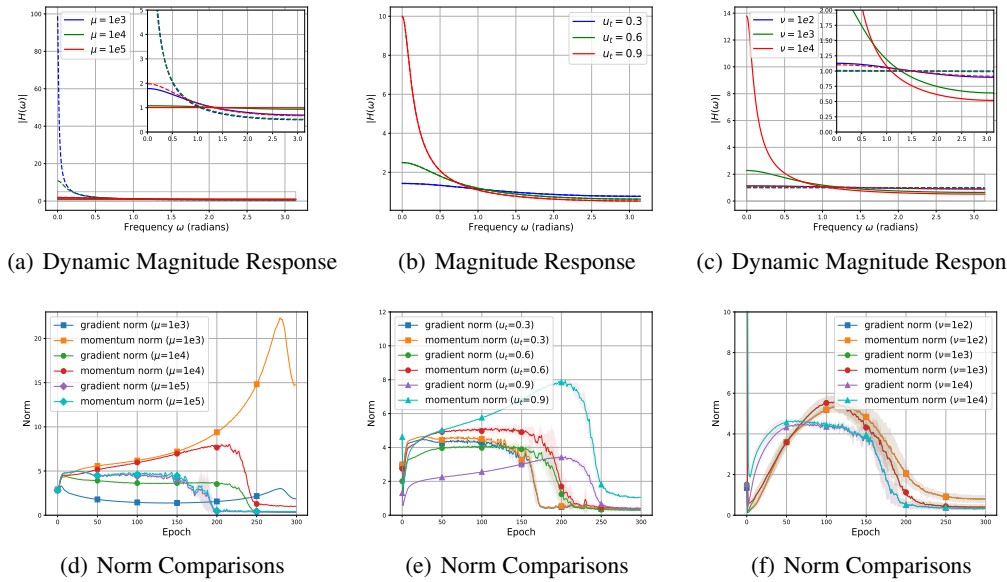

(a) Dynamic Magnitude Response  (b) Magnitude Response  (c) Dynamic Magnitude Response

(d) Norm Comparisons  (e) Norm Comparisons  (f) Norm Comparisons

Figure 3: (**Up**) Analysis of the (dynamic) magnitude responses in the early and late training stages for Standard-SGDM with low-pass gain momentum defined in Equation 10. The *solid lines* denote the magnitude responses in the *early stages*, and the *dashed lines* denote the magnitude responses in the *late stages*. (**Down**) The comparison between the gradient norms and momentum norms for Standard-SGDM with low-pass gain momentum. Left Column: increasing sequence. Middle Column: fixed sequence. Right Column: decreasing sequence.

magnitude responses over time. A large $\mu$ corresponds to the slow shifts. Decreasing $u_t$ reverses the trend, heavily amplifying low-frequency components early and relaxing this effect over time. Fixed $u_t$ maintains constant filters, in which larger $u_t$ amplifies low-frequency components more aggressively. Figure 3(d) - 3(f) demonstrate larger momentum norms compared to gradient norms, indicating the amplification effects in gain filters. Larger $u_t, \nu$ and smaller $\mu$ lead to more reduced momentum norms, which validates the time-variant filtering characteristics of orthodox momentum systems. Test results in Table 2 indicate that increasing $u_t$ with appropriate $\mu$ outperforms the scenarios using fixed and decreasing sequences of $u_t$. We also find that smaller $\nu$ yields worse accuracy in test sets. From these results, we conclude that amplifying low-frequency gradient components and properly attenuating high-frequency ones, improves test set performance.

**Example 4: High-Pass Gain Momentum.** High-pass gain momentum mirrors the dynamic magnitude response behavior of low-pass gain systems (see Figure 7 in App C.2). Increasing $u_t$ gradually amplifies high-frequency gradient components and attenuates low-frequency ones. Decreasing $u_t$ reverses this pattern, heavily amplifying high-frequency components early on. Fixed constructions more aggressively amplify high-frequency components for larger $u_t$. The comparison of gradient norms and momentum norms can be found in Appendix C.2. Test accuracy in Table 2 shows that fixed constructions with larger $u_t$ and decreasing $u_t$ with larger $\nu$ perform worse. These findings confirm that amplifying high-frequency gradients in training might be undesirable.

From Examples 3 and 4, we empirically verify that proper amplification in unorthodox momentum systems can improve model performance, particularly when amplifying low-frequency gradient components.

Table 2: Top-1 ACC. (%) comparisons of different momentum coefficient strategies of unorthodox momentum systems of ResNet50 on CIFAR-100.

| Parameters | Increasing Factor ($\mu$) | | | Fixed Value ($u_t$) | | | Decreasing Factor ($\nu$) | | |
|---|---|---|---|---|---|---|---|---|---|
| | 1k | 10k | 100k | 0.3 | 0.6 | 0.9 | 100 | 1k | 10k |
| Low-Pass Gain | $76.10_{0.14}$ | $80.48_{0.03}$ | $78.02_{0.03}$ | $78.01_{0.04}$ | $79.51_{0.15}$ | $79.71_{0.25}$ | $70.37_{0.67}$ | $71.53_{0.62}$ | $76.18_{0.38}$ |
| High-Pass Gain | $75.47_{0.21}$ | $74.54_{0.16}$ | $75.97_{0.27}$ | $75.68_{0.18}$ | $74.56_{0.09}$ | $73.77_{0.18}$ | $76.41_{0.41}$ | $74.00_{0.26}$ | $68.90_{0.82}$ |

## 3.3 DISCUSSION

The differences in norm comparison and test accuracy between orthodox and unorthodox momentum systems validate the distinctions between EMA-SGDM and Standard-SGDM. While EMA-SGDM possesses attenuating filter effects, Standard-SGDM can both amplify and attenuate different frequency gradient components. Moreover, our findings indicate that with appropriate momentum coefficients, Standard-SGDM consistently outperforms EMA-SGDM, showing the advantages of decoupling momentum coefficients, which answers Question 1.

Regarding Question 2, the test results show that decoupled momentum coefficients with a properly increasing $u_t$ and fixed $v_t$ can achieve better performance. In particular, our empirical findings reveal the following insights in training convolutional neural networks (CNNs): **(1)** high-frequency gradient components are undesired in the late stages of training; **(2)** preserving the original gradient in the early stages leads to improved test set accuracy; **(3)** gradually amplifying low-frequency gradient components enhances performance. Furthermore, we find that these insights are also adaptable in various learning areas (see Section 5). Based on these insights, it may be possible to design a more effective optimizer by appropriately adjusting the momentum coefficients.

## 4 FREQUENCY-BASED OPTIMIZER

As suggested by our frequency domain analysis framework, achieving better test performance is equivalent to finding an appropriate dynamic filter-changing pattern for momentum systems. Based on this idea, we propose FSGDM, a heuristic optimizer that dynamically adjusts momentum filtering characteristics. Furthermore, to explore the potential optimal strategies of our proposed FSGDM based on the findings in Section 3.3, several sets of experiments in various deep-learning tasks are conducted.

## 4.1 FREQUENCY STOCHASTIC GRADIENT DESCENT WITH MOMENTUM

---
**Algorithm 1:** FSGDM

**Input:** $\Sigma, c, v, N$;
**Initialization:** $m_0, \mu = c\Sigma,$
 $\delta = \Sigma/N$;
**for** *each* $t = 1, 2, \ldots$ **do**
 $\quad g_t = \nabla\mathcal{L}_t(x_{t-1}, \zeta_{t-1})$;
 $\quad u(t) = \frac{t}{t+\mu}, \quad u_t = u(\lfloor t/\delta \rfloor \times \delta)$;
 $\quad m_t = u_t m_{t-1} + v g_t$;
 $\quad x_t = x_{t-1} - \alpha_t m_t$;
**end**

---

Generally, determining the *best* optimization strategy by tuning $u_t$ and $v_t$ according to our frequency domain analysis is challenging. In the field of signal processing, how to select the best filters for different problems is still an open problem. However, we can design a *better* optimizer based on the findings in Section 3.3. Still, there are infinite dynamic magnitude responses that can meet the requirements of the aforementioned findings. Based on Occam's Razor principle, we provide a minimalist form of our proposed optimizer in Algorithm 1, where $\Sigma$ is the total gradient update steps in the whole training process determined by the epoch number and the size of the dataset, $c$ is a scaling factor, $\mathcal{L}_t : \mathbb{R}^d \to \mathbb{R}$ is the loss for the $t$-th step, $\zeta_{t-1}$ denotes a minibatch drawn from the training data, and $N$ is the number of stages. $\mu$ and $v$ are adjustable parameters that dominate the filtering characteristic of FSGDM. Moreover, since $\mu$ is a function of $\Sigma$, the dynamic magnitude response can be inherited when $\Sigma$ varies. In particular, we have the following proposition.

**Proposition 1.** *By fixing the number of stages $N$ and the scaling factor $c$, the dynamic magnitude response of Algorithm 1 keeps invariant with respect to changes in the total number of training steps.*

The proof of Proposition 1 is deferred in Appendix B.3. By this, we show that the dynamic magnitude response of a well-performed FSGDM can be adaptable to various tasks. In the following subsection, we explore the optimal scaling factor $c$ and momentum coefficient $v$ for FSGDM.

## 4.2 EMPIRICAL EXPLORATION OF OPTIMAL SETTINGS FOR FSGDM

As discussed in Section 3, different choices of $c$ and $v$ can significantly affect the filtering characteristics of FSGDM. To understand their impact on optimization performance and to identify optimal parameter settings, we conduct a comprehensive empirical study.

Specifically, we empirically explore the optimal parameter selection of FSGDM across three different image classification tasks by first sweeping $c$ and $v$ within the ranges of $(0, 1)$ and $[0.5, 3]$, respectively. Specifically, we conduct three sets of experiments using the same codebase (See Appendix D for more training details): (1) training ResNet18 for 100 epochs on CIFAR-10, (2) training ResNet34 for 100 epochs on Tiny-ImageNet (Le & Yang, 2015), and (3) training ResNet50 for 300 epochs on CIFAR-100. We also explore the optimal parameter selection on one natural language processing task in Appendix C.7. By finding the parameter selections with better test performance in different tasks, we try to empirically summarize the law of optimal parameter selection.

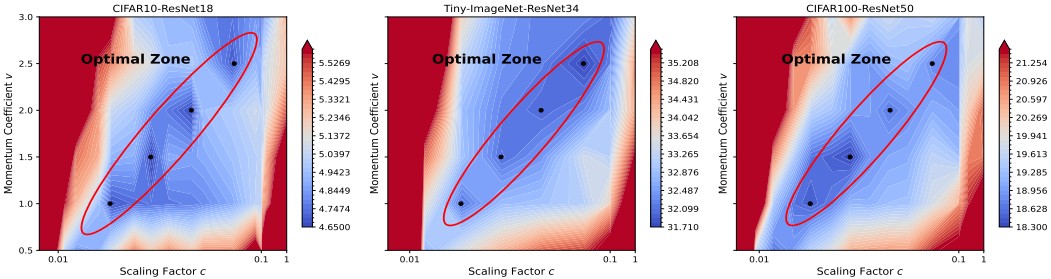

Figure 4: The Top-1 test errors of training ResNet18 on CIFAR-10, ResNet34 on Tiny-ImageNet and ResNet50 on CIFAR-100. The results show that the optimal parameter selections across these three training settings exhibit a high similarity. The black points denote the parameter selections with better test performance. The optimal zone of the parameter selection is circled in red.

The results in Figure 4 show that there exists an optimal zone where relatively better test accuracy results can be achieved. When the momentum coefficient $v$ is fixed, the test accuracy shows an initial increase followed by a decline as the scaling factor $c$ increases. In Appendix C.8, we plot the magnitude responses and the test accuracy results of the black points in Figure 4 and find that these parameter selections have similar dynamic magnitude responses and test accuracy curves. Thus, we assume the parameter selections with similar dynamic magnitude responses will lead to close performance. More discussions are in Appendix C.8.

## 5 EXPERIMENTS

To verify the generalization of the proposed FSGDM, we perform a large-scale comparison across vision classification tasks, natural language processing (NLP) tasks, and reinforcement learning (RL) tasks. We compare the test performance of FSGDM and conventional SGD-based momentum optimizers, including Standard-SGDM and EMA-SGDM. We set $u_t = 0.9, v_t = 1$ for Standard-SGDM, and $u_t = 0.9$ for EMA-SGDM, which are the common momentum coefficient selections in training neural networks. For a fair comparison and convenience, we set $c = 0.033, v = 1$, which is one of the black points in the optimal zone in Figure 4, for FSGDM. Note that other combinations of $c$ and $v$ in the optimal zone can also be selected. For the other adjustable parameters in Algorithm 1, we set N to 300 as mentioned at the end of Section 2.1, and set $\Sigma$ as the number of total training steps. Notably, since our focus is on comparing the performance of different optimizers, we do not fine-tune every parameter for each model but use the same hyperparameters across all models for convenience. See Appendix D for more experimental details.

Table 3: Performance on Image Classification Experiments

| Dataset | CIFAR-10 | | CIFAR-100 | | Tiny-ImageNet | | ImageNet |
|---|---|---|---|---|---|---|---|
| Model | VGG16 | ResNet18 | ResNet50 | DenseNet121 | ResNet34 | MobileNet | ResNet50 |
| EMA-SGDM | $93.71_{0.07}$ | $94.19_{0.07}$ | $76.84_{0.06}$ | $76.18_{0.23}$ | $62.28_{0.17}$ | $55.00_{0.10}$ | $74.24_{0.04}$ |
| Standard-SGDM | $94.08_{0.07}$ | $95.57_{0.06}$ | $79.71_{0.25}$ | $80.49_{0.09}$ | $67.51_{0.08}$ | $58.31_{0.20}$ | $76.66_{0.09}$ |
| **FSGDM** | $\mathbf{94.19}_{0.07}$ | $\mathbf{95.66}_{0.07}$ | $\mathbf{81.44}_{0.06}$ | $\mathbf{81.14}_{0.05}$ | $\mathbf{67.74}_{0.06}$ | $\mathbf{59.61}_{0.11}$ | $\mathbf{76.91}_{0.05}$ |

**Image Classification.** We perform four sets of experiments with different datasets in computer vision tasks and use various CNN architectures for training them. Specifically, we select: (a) VGG16 and ResNet18 for CIFAR-10; (b) ResNet50 and DenseNet121 (Huang et al., 2017) for CIFAR-100; (c) ResNet34 and MobileNet (Howard, 2017) for Tiny-ImageNet; (d) ResNet50 for ILSVRC 2012

ImageNet Russakovsky et al. (2015). For each task, we report the mean and standard error (as subscripts) of test accuracy for 3 runs with random seeds from 0-2. The results in Table 3 show that our FSGDM consistently achieves better test set performance. Additionally, we can observe that Standard-SGDM steadily outperforms EMA-SGDM, which aligns with our discoveries in Section 3.3.

**Natural Language Processing.** We conduct experiments on the IWSLT14 German-English translation task (Cettolo et al., 2014) to represent NLP tasks, a widely used benchmark in the community. Specifically, we train six different models encompassing a variety of architectures: two convolution-based models, FConv (Gehring et al., 2017) and LightConv (Wu et al., 2019); two LSTM-based models, vanilla LSTM (Hochreiter & Schmidhuber, 1997) and LSTM-W (Wiseman & Rush, 2016); and two Transformer-based models (Vaswani et al., 2017) with different sizes, Transformer-tiny and Transformer. Model performance is reported using BLEU scores, where higher scores indicate better performance, and we summarize all results in Table 4. Compared with the baseline optimizers, FSGDM outperforms all others in this task across six different models. This shows the effectiveness of our optimizer in improving translation quality. Moreover, the consistent improvement highlights the robustness of FSGDM and its ability to generalize across different neural network structures in natural language processing tasks.

Table 4: Performance on IWSLT14 Dataset

| Model | FConv | LightConv | LSTM | LSTM-W | Transformer-tiny | Transformer |
|---|---|---|---|---|---|---|
| EMA-SGDM | $13.97_{0.01}$ | $10.56_{0.01}$ | $4.99_{0.01}$ | $1.20_{0.07}$ | $5.17_{0.01}$ | $6.27_{0.01}$ |
| Standard-SGDM | $27.41_{0.02}$ | $33.05_{0.04}$ | $28.12_{0.06}$ | $24.66_{0.06}$ | $18.16_{0.03}$ | $31.50_{0.05}$ |
| **FSGDM** | $\mathbf{28.30_{0.01}}$ | $\mathbf{33.44_{0.02}}$ | $\mathbf{29.27_{0.02}}$ | $\mathbf{27.41_{0.03}}$ | $\mathbf{19.94_{0.07}}$ | $\mathbf{32.40_{0.05}}$ |

Figure 5: The reward curves of EMA-, Standard-SGDM, and FSGDM on three MuJoCo tasks.

**Reinforcement Learning.** We evaluate FSGDM on PPO (Schulman et al., 2017), one of the most popular policy gradient methods in reinforcement learning. We replace the default Adam optimizer (Kingma & Ba, 2014) in PPO with FSGDM, Standard-SGDM, and EMA-SGDM. We test the three optimizers on Walked2d-v4, HalfCheetah-v4, and Ant-V4, which are continuous control environments simulated by the standard and widely-used engine, MuJoCo (Todorov et al., 2012). Following standard evaluation, we run each game under 10 random seeds (range from 0-9) and test the performance for 10 episodes every 30,000 steps. All experiments are conducted using the Tianshou codebase (Weng et al., 2022), a widely known RL framework. Figure 5 presents the results on three tasks, where the solid line represents the average episode rewards during evaluation, and the shaded region indicates the 75% confidence interval. It can be easily observed that on three test games, our FSGDM achieves higher rewards than Standard-SGDM and EMA-SGDM.

## 6 CONCLUSIONS

This paper proposes a frequency domain analysis framework for the momentum method. Based on the proposed framework, we find that different selections of momentum coefficients correspond to different filter characteristics of the momentum methods. Performance will be significantly different under different time-variant momentum coefficients. Furthermore, we develop a heuristic optimizer named FSGDM which outperforms the conventional SGD-based momentum optimizers in various learning tasks. Future work may explore the *best* filtering strategy for all general scenarios and extend the frequency domain analysis framework to other optimizers such as Adam.

ACKNOWLEDGMENTS

We would like to especially thank Prof. K. C. Ho for many enlightening discussions. The work of Xianliang Li and Sheng Xu was supported by the National Natural Science Foundation of China (62273327) and the Shenzhen Science and Technology Program (KCXFZ20211020165003005). The work of Linlong Wu was supported by Luxembourg FNR CORE METSA project C22/IS/17391632.

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

## A  PHASE RESPONSE

The phase response of the momentum system in the $k$-th stage can be written as,

$$\arg(H_k(\omega)) = \arg(v_k) - \tan^{-1}\left(\frac{u_k \sin\omega}{1 - u_k \cos\omega}\right), \tag{11}$$

where $\arg(\cdot)$ is the argument operator. For any real value $v_k$, $\arg(v_k) = 0$ if $v_k > 0$ and $\arg(v_k) = \pi$ if $v_k < 0$; for any $\omega \in [0, \pi]$ and $u_k \in (-1, 1)$, $\tan^{-1}(u_k \sin\omega/(1 - u_k \cos\omega)) \in (-\frac{\pi}{2}, \frac{\pi}{2})$. The phase response describes the phase-shifting effect of the momentum system at different frequencies. In the context of gradient-based optimization, the phase shift indicates a change in the optimization direction. Therefore, when $v_k < 0$, the phase shift of the momentum adds up an extra $\pi$ rad on the shifted direction, indicating that the direction of the update is greatly reversed, which can lead to oscillations, instability, or divergence in the optimization process. Thus, it is necessary to select a positive $v_k$ when applying momentum methods.

## B  ADDITIONAL DERIVATIONS AND PROOF

### B.1  DERIVATION OF EQUATION 8

$$\begin{aligned}
|H_k(\omega)| &= \sqrt{H_k(\omega)H_k^\dagger(\omega)} \\
&= \sqrt{\frac{v_k}{1 - u_k e^{-j\omega}} \cdot \frac{v_k}{1 - u_k e^{j\omega}}} \\
&= \sqrt{\frac{v_k^2}{1 - u_k e^{-j\omega} - u_k e^{j\omega} + u_k^2 e^{-j\omega} e^{j\omega}}} \\
&= \sqrt{\frac{v_k^2}{1 - u_k(\cos\omega - j\sin\omega) - u_k(\cos\omega + j\sin\omega) + u_k^2(\cos^2\omega + \sin^2\omega)}} \\
&= \sqrt{\frac{v_k^2}{1 - 2u_k \cos\omega + u_k^2}} \\
&= \frac{|v_k|}{\sqrt{1 - 2u_k \cos\omega + u_k^2}}
\end{aligned}$$

### B.2  DERIVATION OF EQUATION 11

$$\begin{aligned}
\arg(H_k(\omega)) &= \arg(v_k) - \arg(1 - u_k e^{-j\omega}) \\
&= \arg(v_k) - \arg\left((1 - u_k \cos\omega) + j(u_k \sin\omega)\right) \\
&= \arg(v_k) - \tan^{-1}\left(\frac{u_k \sin\omega}{1 - u_k \cos\omega}\right)
\end{aligned}$$

### B.3  PROOF OF PROPOSITION 1

According to Algorithm 1, the momentum coefficient in the $k$-th stage ($k = 1, 2, \cdots, N$) is

$$u_k = \frac{(k-1)\delta}{(k-1)\delta + \mu} = \frac{(k-1)\delta}{(k-1)\delta + \Sigma/c} = \frac{(k-1)\delta}{(k-1)\delta + cN\delta} = \frac{k-1}{k-1+cN}. \tag{12}$$

This guarantees that the number of training steps, which may be different when choosing other training strategies or changing datasets, is independent of $u_k$ when the scaling factor $c$ and the number of stages $N$ are already determined.

## C  ADDITIONAL EXPERIMENTS

In this section, we present several supplementary experiments. The detailed experimental settings are shown in Appendix D.

### C.1  DYNAMIC SEQUENCE CONSTRUCTION

There are infinite increasing or decreasing sequences. In this part, we compare the test set performance of the sequence mentioned in Equation 4 with four other dynamic sequences. Specifically, we compare with the following four dynamic increasing sequences within Algorithm 1:

$$\text{Linear:} \quad u(t) = a_1 t;$$
$$\text{Exponential:} \quad u(t) = 1 - e^{-a_2 t};$$
$$\text{Sine:} \quad u(t) = \sin(a_3 t);$$
$$\text{Logarithmic:} \quad u(t) = \ln(a_4 t);$$

where $a_1$ to $a_4$ are scaling coefficients. For a fair comparison, we adjust the coefficients to keep the $u_t$ of all sequences unchanged in the beginning and ending stages. To make other types of sequences unique, we keep the $u_t$ of different dynamic sequences nearly unchanged in the beginning and ending stages. Table 5 displays their test accuracy results after 300 epochs of training on CIFAR-100 using ResNet50. We ran each experiment under 3 different random seeds (0, 1, 2). Clearly, the dynamic sequence we use in Equation 4 shows its superiority over other constructions.

Table 5: Top-1 ACC. (%) comparisons of using linear, exponential, sine, logarithmic, and our sequences when adopting FSGDM.

| Dynamic Sequence Type | Ours | Linear | Exponential | Sine | Logarithmic |
|---|---|---|---|---|---|
| ACC-1 (%) | $\mathbf{81.44}_{0.06}$ | $78.24_{0.24}$ | $80.38_{0.04}$ | $78.76_{0.29}$ | $78.70_{0.09}$ |

Specifically, $(a_1, a_2, a_3, a_4) = (8.271 \times 10^{-6}, 3.793 \times 10^{-5}, 1.125 \times 10^{-5}, 1.394 \times 10^{-5})$.

### C.2  ADDITIONAL FIGURES OF HIGH-PASS MOMENTUM SYSTEMS ON CIFAR-100

This subsection provides the figures of the dynamic magnitude responses and norm of high-pass (gain) momentum systems mentioned in Section 3. Figure 6 and Figure 7 show the magnitude responses and norm comparisons of high-pass and high-pass gain momentum systems, respectively. The high-pass (gain) momentum systems preserve or even amplify rapidly fluctuating gradient components, leading to sharp oscillations in gradient norm curves and momentum norm curves across iterations.

### C.3  ADDITIONAL EXPERIMENTS OF VGG16 ON CIFAR-10

In this subsection, we provide experiments of training VGG16 on CIFAR-10. The experimental settings follow Section 3 and Appendix D. From the test accuracy in Table 6 and Table 7, we observe that the test performances and norm comparisons in different momentum methods in training VGG16 on CIFAR-10 are similar to those in training ResNet50 on CIFAR-100. This similarity implies that the empirical findings in Section 3 are applicable to various CNNs.

Table 6: Comparison of Top-1 Accuracy (%) among different momentum coefficient methods in orthodox momentum systems using VGG16 on CIFAR-10.

| Parameters | Increasing Factor ($\mu$) | | | Fixed Value ($u_t$) | | | Decreasing Factor ($\nu$) | | |
|---|---|---|---|---|---|---|---|---|---|
| | 1k | 10k | 100k | 0.3 | 0.6 | 0.9 | 100 | 1k | 10k |
| Low-Pass | $93.80_{0.05}$ | $93.78_{0.12}$ | $93.79_{0.09}$ | $93.68_{0.18}$ | $93.64_{0.08}$ | $93.71_{0.07}$ | $92.33_{0.04}$ | $90.89_{0.11}$ | $90.56_{0.19}$ |
| High-Pass | $90.02_{0.05}$ | $92.64_{0.09}$ | $93.41_{0.01}$ | $93.52_{0.16}$ | $92.71_{0.07}$ | $90.32_{0.07}$ | $93.86_{0.09}$ | $93.73_{0.08}$ | $93.38_{0.09}$ |

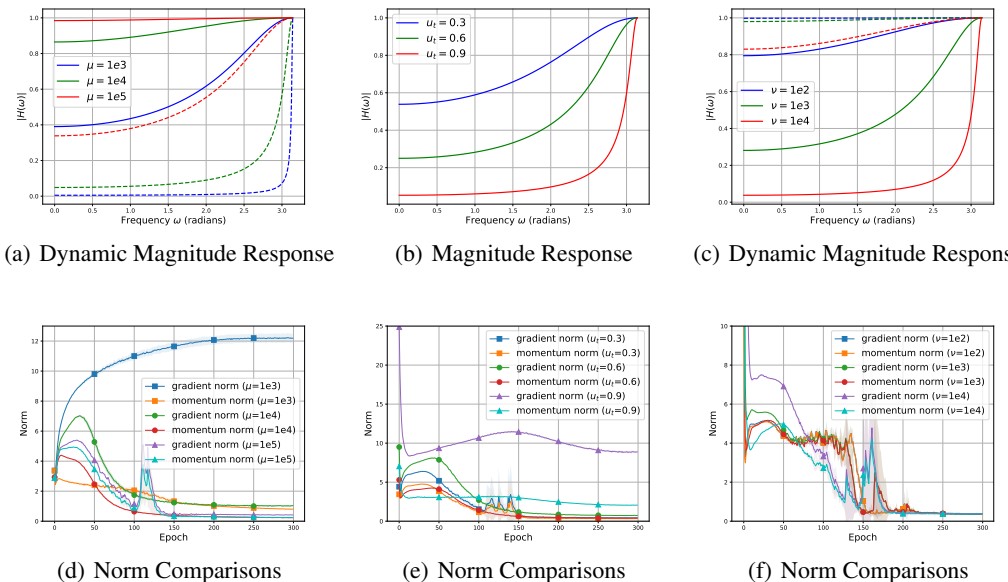

Figure 6: (**Up**) Analysis of the (dynamic) magnitude responses in the early and late training stages for EMA-SGDM with high-pass momentum defined in Equation 9. The *solid lines* denote the magnitude responses in the *early stages*, and the *dashed lines* denote the magnitude responses in the *late stages*. (**Down**) The comparison between the gradient norms and momentum norms for EMA-SGDM with high-pass momentum. Left Column: increasing sequence. Middle Column: fixed sequence. Right Column: decreasing sequence.

Table 7: Comparison of Top-1 Accuracy (%) among different momentum coefficient methods in unorthodox momentum systems using VGG16 on CIFAR-10.

| Parameters | Increasing Factor ($\mu$) | | | Fixed Value ($u_t$) | | | Decreasing Factor ($\nu$) | | |
|---|---|---|---|---|---|---|---|---|---|
| | 1k | 10k | 100k | 0.3 | 0.6 | 0.9 | 100 | 1k | 10k |
| Low-Pass Gain | $84.01_{0.13}$ | $94.19_{0.07}$ | $93.85_{0.07}$ | $93.86_{0.11}$ | $93.98_{0.09}$ | $94.08_{0.07}$ | $92.00_{0.05}$ | $92.27_{0.12}$ | $92.97_{0.23}$ |
| High-Pass Gain | $93.34_{0.03}$ | $93.56_{0.06}$ | $93.79_{0.13}$ | $93.71_{0.11}$ | $93.46_{0.06}$ | $93.33_{0.02}$ | $93.79_{0.07}$ | $93.33_{0.12}$ | $93.05_{0.08}$ |

## C.4 THE EARLY STAGES OF TRAINING

This subsection focuses on the test performance affected by the momentum coefficients in the very early training stages. We plot the test accuracy curves for the first 10 epochs of different momentum systems in Section 3 and study the early behaviors of different momentum systems.

Figure 8 demonstrates the early test accuracy curves of different momentum coefficient methods. For orthodox momentum systems, preserving the original gradient (i.e., all-pass momentum system, low-pass momentum system with an increasing $u_t$, and high-pass momentum system with an increasing $u_t$) or attenuating high-frequency gradient components(i.e., static low-pass momentum system with $u_t = 0.9$) results in better initial performance, while greatly attenuating high-frequency gradient components (i.e., low-pass momentum system with a decreasing $u_t$) or attenuating low-pass components (i.e., static high-pass and high-pass momentum system with a decreasing $u_t$) lead to bad test performance at the beginning.

On the other hand, for unorthodox momentum systems, preserving the original gradient (i.e., all-pass momentum system, low-pass gain momentum system with an increasing $u_t$, and high-pass gain momentum system with an increasing $u_t$) can achieve better early performance, while greatly amplifying high-frequency gradient components (i.e., static high-pass gain momentum system and high-pass gain momentum system with a decreasing $u_t$) leads to bad initial accuracy results.

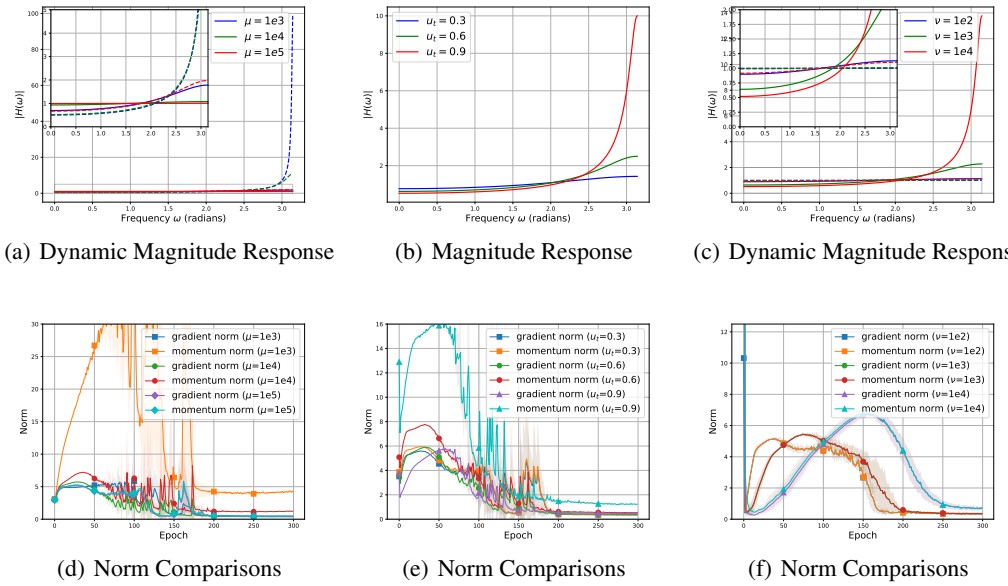

(a) Dynamic Magnitude Response     (b) Magnitude Response     (c) Dynamic Magnitude Response

(d) Norm Comparisons     (e) Norm Comparisons     (f) Norm Comparisons

Figure 7: (**Up**) Analysis of the (dynamic) magnitude responses in the early and late training stages for Standard-SGDM with high-pass gain momentum defined in Equation 10. The *solid lines* denote the magnitude responses in the *early stages*, and the *dashed lines* denote the magnitude responses in the *late stages*. (**Down**) The comparison between the gradient norms and momentum norms for Standard-SGDM with high-pass gain momentum. Left Column: increasing sequence. Middle Column: fixed sequence. Right Column: decreasing sequence.

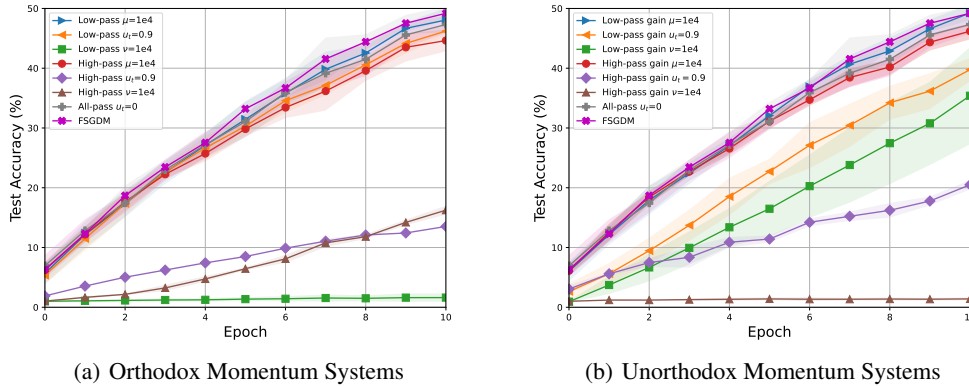

(a) Orthodox Momentum Systems     (b) Unorthodox Momentum Systems

Figure 8: The first 10 epochs of the test accuracy curves with different momentum coefficient methods. We choose $10^4$ for both increasing and decreasing factors ($\mu$ and $\nu$) in dynamic momentum systems and $u_t = 0.9$ for static momentum coefficient.

These observations significantly validate that preserving the original gradient in early stages enhances test performance, which matches the findings in Section 3. Additionally, our proposed FSGDM retains the all-pass characteristic and possesses the same quick start property in test accuracy curves.

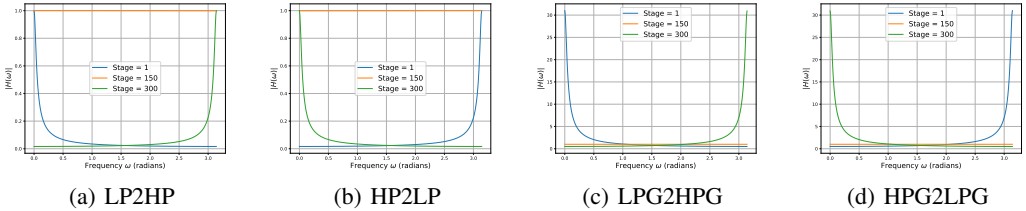

Figure 9: The magnitude response curves of Stage $1, 150, 300$ in different momentum systems.

## C.5 COMPARISON WITH SPECIAL MOMENTUM SYSTEMS

In this subsection, we investigate the test performance of the following four types of momentum systems: 1) low-pass to high-pass momentum system (LP2HP); 2) high-pass to low-pass momentum system (HP2LP); 3) low-pass gain to high-pass gain momentum system (LPG2HPG); 4) high-pass gain to low-pass gain momentum system (HPG2LPG). Their dynamic magnitude responses are shown in Figure 9. Note that the maximum values $|H(\omega)|$ of these four systems are the same as the default setting in FSGDM. We run each experiment under 3 different random seeds (0-2). Table 8 displays the test accuracy results of four types of momentum systems and FSGDM. Our proposed FSGDM outperforms all four special momentum systems. Specifically, the test accuracy of the momentum systems shifting from high-pass to low-pass is better than that shifting from low-pass to high-pass. This indicates that compared to the low-frequency gradient components, high-frequency components are more undesired in the late training stages, which supports the finding in Section 3.

Table 8: Comparison of Top-1 Accuracy (%) among the low-pass to high-pass, high-pass to low-pass, low-pass gain to high-pass gain, high-pass gain to low-pass gain momentum systems and FSGDM.

| Dynamic Magnitude Response | FSGDM | LP2HP | HP2LP | LPG2HPG | HPG2LPG |
|---|---|---|---|---|---|
| ACC-1 (%) | $\mathbf{81.44}_{0.06}$ | $74.77_{0.21}$ | $77.00_{0.13}$ | $72.60_{0.58}$ | $78.91_{0.25}$ |

## C.6 TRAINING WITH EXTREME MOMENTUM COEFFICIENTS

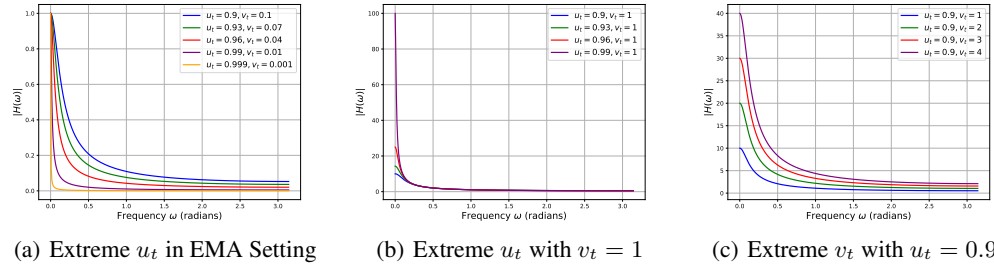

(a) Extreme $u_t$ in EMA Setting      (b) Extreme $u_t$ with $v_t = 1$      (c) Extreme $v_t$ with $u_t = 0.9$

Figure 10: The magnitude responses of different $u_t$ and $v_t$ with extreme value ranges. (a): EMA-SGDM; (b), (c): Standard-SGDM.

Why do researchers usually choose $u_t = 0.9$ or $v_t = 1$ instead of larger values? From the frequency domain perspective, we discover that 1) when $u_t$ is extremely close to 1 in EMA-SGDM, the momentum system will behave like a super narrow low-pass filter, with an extreme reduction in most of the high-frequency gradient components; 2) when $u_t$ is extremely close to 1 in Standard-SGDM, the momentum system will behave like a super narrow low-pass gain filter, with a reduction in high-frequency gradient components and high amplification in a narrow band of low-frequency gradient

components; 3) when $v_t$ is larger than $1$ in Standard-SGDM, the attenuation of high-frequency gradient components is then reduced. We speculate that all these poor filtering characteristics of the momentum systems will lead to bad test performance. Figure 10 displays the magnitude response of these three situations. As shown in Figure 11, the test performance results validate our previous speculations and support our frequency domain analysis framework.

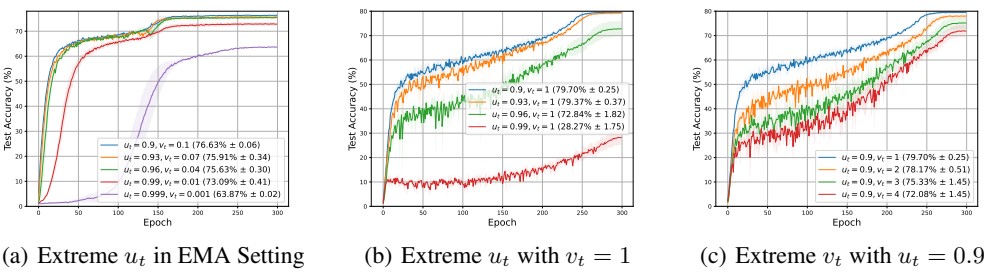

(a) Extreme $u_t$ in EMA Setting     (b) Extreme $u_t$ with $v_t = 1$     (c) Extreme $v_t$ with $u_t = 0.9$

Figure 11: The test accuracy curves of different $u_t$ and $v_t$ in extreme value ranges. (a): EMA-SGDM; (b), (c): Standard-SGDM.

### C.7 ADDITIONAL EXPLORATION OF OPTIMAL SETTINGS FOR NLP TASKS

In this subsection, we provide experiments that explore the optimal parameter selection of FSGDM for the IWSLT14 translation task by training LSTM-W and Transformer-tiny. The experimental settings follow Section 5 and Appendix D.

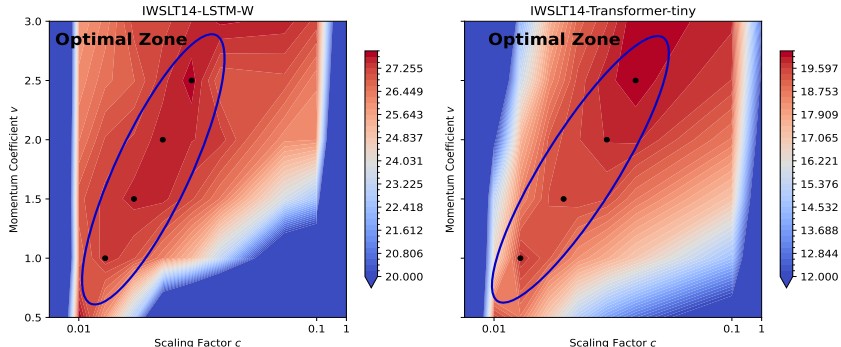

Figure 12: The BLEU scores of training LSTM-W and Transformer-tiny on IWSLT14 German-English translation task. The results show that the optimal parameter selections across these two training settings exhibit a high similarity. The black points denote the parameter selections with better test performance. The optimal zone of the parameter selection is circled in blue.

The results in Figure 12 indicate that similar optimal zones can be observed on the NLP task. When the momentum coefficient $v$ is fixed, the BLEU score shows an initial increase followed by a decline as the scaling factor $c$ increases, which is highly consistent with the results in Section 4.2. In addition, we find that the empirical insights discussed in Section 3.3 are also applicable to various deep learning models beyond CNNs, as well as NLP tasks.

### C.8 OPTIMAL ZONE OF FSGDM

In this subsection, we go deeper into the optimal zone. We suspect that the similarity of the dynamic magnitude responses may lead to close test set performance. The dynamic magnitude responses of the black points with different parameters in the optimal zone (Figure 4) are shown in Figure 13. We train ResNet50 on CIFAR-100 and visualize the training losses and the test accuracy curves of different points in the optimal zone. The results are shown in Figure 14.

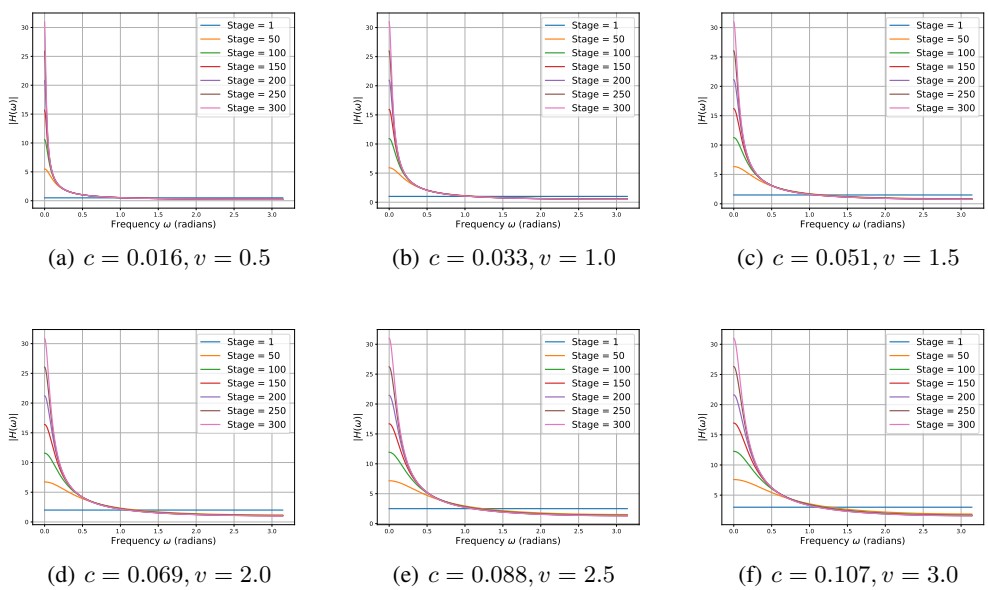

Figure 13: The dynamic magnitude responses of the black points in the optimal zone.

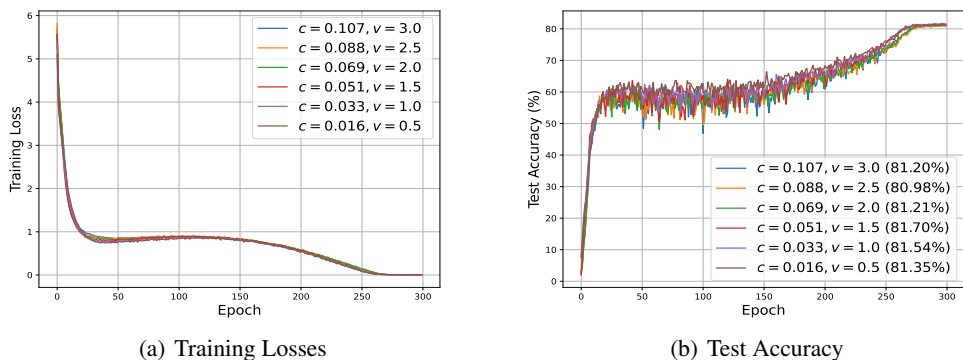

(a) Training Losses        (b) Test Accuracy

Figure 14: The training losses and test accuracy of different parameter settings in the optimal zone.

From the training loss and test accuracy curves, we find that the optimization processes of different black points in the optimal zone resemble each other. According to the existing parameter settings of the black points, one can find that the mathematical relationship between $c$ and $v$ in training ResNet50 on CIFAR-100 is approximately $\frac{30.992}{v} \approx 1 + \frac{1}{c}$ [4].

## C.9 ABLATION STUDY ON DIFFERENT BATCH SIZE

This subsection provides the ResNet50 training experiments on CIFAR-100 with different batch size settings. We compare the Top-1 accuracy of the test set by using our FSGDM with $c = 0.033, v = 1$, Standard-SGDM with $u_t = 0.9, v_t = 1$, and EMA-SGDM with $u_t = 0.9$, as shown in Table 9. The test results show that our FSGDM consistently outperforms popular conventional SGD-based momentum optimizers.

---

[4] This relationship can be better approximated and generalized with continued experimentations across diverse tasks.

Table 9: Comparison of Top-1 Accuracy (%) among the FSGDM, Standard-SGDM, and EMA-SGDM with different batch size settings.

| Batch size | 64 | 128 | 256 |
|---|---|---|---|
| EMA-SGDM | $79.42_{0.11}$ | $76.84_{0.06}$ | $69.03_{0.39}$ |
| Standard-SGDM | $79.55_{0.13}$ | $79.71_{0.25}$ | $78.96_{0.33}$ |
| FSGDM | $\mathbf{80.92}_{0.13}$ | $\mathbf{81.44}_{0.06}$ | $\mathbf{80.34}_{0.01}$ |

# D EXPERIMENTAL SETTINGS

## D.1 TRAINING SETTINGS FOR VISION CLASSIFICATION TASKS

We use custom training code based on the PyTorch tutorial code for all our visual classification experiments (including the experiments in Section 3, Section 4.2 and Section 5) We choose the CosineAnnealingLR (Loshchilov & Hutter, 2016) as our training scheduler. Additionally, we set the learning rate as $1 \times 10^{-1}$ for all experiments, while the weight decay is set as $5 \times 10^{-4}$ for experiments on CIFAR-10, CIFAR-100, and Tiny-ImageNet, and $1 \times 10^{-1}$ for ImageNet. All models we used are simply following their paper's original architecture, and adopt the weight initialization introduced by He et al. (2015). Additionally, we train 300 epochs for experiments on CIFAR-10 and CIFAR-100 and train 100 epochs for Tiny-ImageNet and ImageNet. We use a 128 batch size for experiments on CIFAR-10, CIFAR-100, and Tiny-ImageNet, and 256 for ImageNet. All experiments are conducted on RTX 4090 or A100 GPUs.

**Data Augmentation.** For experiments on CIFAR-10, CIFAR-100, and Tiny-ImageNet, we adopt PyTorch's RandomCrop, followed by random horizontal flips. Specifically, the random crop size is set to 32x32 for CIFAR-10 and CIFAR-100 and set to 64x64 for Tiny-ImageNet. For experiments on ImageNet, we adopt PyTorch's RandomResizedCrop, cropping to 224x224 followed by random horizontal flips. Test images use a fixed resize to 256x256 followed by a center crop to 224x224. At last, a data normalization is adopted to input images.

## D.2 TRAINING SETTINGS FOR NATURAL LANGUAGE PROCESSING TASKS

All models used in our experiments are directly adopted from the FairSeq [5] framework. We retain the original architecture of each model and train all models for 100 epochs using a single NVIDIA RTX 4090 GPU. We set the maximum batch size to 4,096 tokens and apply gradient clipping with a threshold of 0.1. The baseline learning rate is set to 0.25, and for the optimizer, we use a weight decay of 0.0001.

## D.3 TRAINING SETTINGS FOR REINFORCEMENT LEARNING TASKS

For the experiments in RL tasks, we do not make any changes except for replacing the original Adam optimizer with Standard-SGDM, EMA-SGDM, and our proposed FSGDM. To ensure fairness, we use Tianshou's (Weng et al., 2022) default hyperparameters for PPO training. However, since SGD-based optimizers are highly sensitive to the learning rate, we searched for suitable learning rates across the three games, ultimately setting $10^{-2}$, $10^{-2}$ and $10^{-3}$ for Walker2d-v4, HalfCheetah-v4, and Ant-v4, respectively.

---

[5]https://github.com/facebookresearch/fairseq

# E    CHALLENGES IN THE FREQUENCY DOMAIN ANALYSIS FOR ADAPTIVE OPTIMIZERS

---

**Algorithm 2:** RMSprop

---

**Input** $\beta_2$, $\epsilon$, $v_0$;
**for** *each $t = 1, 2, \ldots$* **do**
$\quad g_t = \nabla \mathcal{L}_t(x_{t-1}, \zeta_{t-1})$;
$\quad v_t = \beta_2 v_{t-1} + (1 - \beta_2)g_t^2$;
$\quad x_t = x_{t-1} - \alpha_t g_t / (\sqrt{v_t} + \epsilon)$;
**end**

---

**Algorithm 3:** Adam

---

**Input** $\beta_1$, $\beta_2$, $\epsilon$, $m_0$, $v_0$;
**for** *each $t = 1, 2, \ldots$* **do**
$\quad g_t = \nabla \mathcal{L}_t(x_{t-1}, \zeta_{t-1})$;
$\quad m_t = \beta_1 m_{t-1} + (1 - \beta_1)g_t$;
$\quad v_t = \beta_2 v_{t-1} + (1 - \beta_2)g_t^2$;
$\quad \widehat{m_t} = \frac{m_t}{1-\beta_1^t}, \widehat{v_t} = \frac{v_t}{1-\beta_2^t}$;
$\quad x_t = x_{t-1} - \alpha_t \widehat{m_t} / (\sqrt{\widehat{v_t}} + \epsilon)$;
**end**

---

In this section, we make a discussion on the potential challenges for the extension of the frequency domain analysis framework to adaptive optimizers like RMSprop and Adam as shown in Algorithm 2 and 3. The first-moment estimate of Adam is in the form of EMA and thus acts as a low-pass filter. However, the second-moment estimate presents additional obstacles for frequency domain analysis in the following ways:

1. The second-moment estimates of Adam and RMSprop involve the squared gradient term $g_t^2$, resulting in nonlinearity that complicates the direct application of the Z-transform.

2. Adam introduces both the first- and second-moment estimates ($m_t$ and $v_t$), and adopts $\widehat{m_t} / (\sqrt{\widehat{v_t}} + \epsilon)$ as the update step. This intricate interaction between $m_t$ and $v_t$ also makes the analysis more challenging.

At this stage, we believe that our argument regarding the three insights discussed in Section 3.3 is also applicable to other optimizers. However, it remains unclear how the different frequency gradient components in the model parameter updates are processed by the Adam optimizer. We anticipate that resolving these issues will provide deeper insight.

