# OpenReview forum: "On the Performance Analysis of Momentum Method: A Frequency Domain Perspective"
_ICLR.cc/2025/Conference — ICLR 2025 Poster_

### Official Review · Reviewer_RGqa · 2024-10-25

**Soundness:** 3
**Presentation:** 3
**Contribution:** 3
**Rating:** 6
**Confidence:** 4

**Summary:**

The paper explores the momentum-based optimization methods commonly used in training neural networks through a frequency domain perspective. It shows a new interpretation of momentum as a time-variant gradient filter, where the value of momentum coefficients determines the filtering of different frequency components of gradients. The analysis shows that high-frequency gradient components decrease performance in the later training stages, while preserving original gradients in the early stages and amplifying low-frequency components gradually improves generalization.

Building on these insights, the paper introduces Frequency Stochastic Gradient Descent with Momentum (FSGDM), a heuristic optimizer that adjusts the momentum coefficients dynamically to enhance training performance. Experimental results on image classification and NLP tasks demonstrate that FSGDM outperforms conventional momentum-based optimizers.

**Strengths:**

Originality:
* The paper explores the impact of momentum on gradient frequency domain. This work is very original and has not been well explored before. The exploration shows that the momentum update is usually a low pass filter which gets more and more narrower low pass filter in the later part of the training.

Quality:
* The derivations based on signal processing principles, particularly the application of Z-transforms, provide a good theoretical foundation. The analysis effectively distinguishes between conventional momentum methods (like Standard-SGDM and EMA-SGDM), clarifying their distinct impacts on gradient components from frequency domain perspective.
* The experiments are comprehensive and well-designed, covering image classification and language translation. The results consistently show that FSGDM outperforms traditional momentum optimizers in both accuracy and convergence speed, demonstrating the practical relevance of the proposed method.

Clarity:
*  The paper follows a clear structure, beginning with the theoretical formulation of the problem, progressing to the design of the FSGDM optimizer, and concluding with comprehensive experimental results.
* The use of visuals, such as plots of frequency response and performance comparisons, enhances understanding. The inclusion of tables summarizing performance across different tasks is also helpful for comparing results at a glance.


Significance:
* By integrating frequency domain insights into momentum adjustment, this work not only advances the understanding of optimization mechanisms but also opens the door for further research.
* The empirical results demonstrate that FSGDM consistently achieves better performance than standard momentum-based optimizers, indicating that the proposed method can be a strong alternative across different neural network task

**Weaknesses:**

Significance:
* While the Frequency perspective of momentum based weight updates is very impressive, the proposed FSGDM (Frequency Stochastic Gradient Descent with Momentum) does not appear to be very significant because of the following 2 reasons
  1. Setting schedules for momentum coefficient and learning rates during training is a very common practice in NN training. In the proposed FSGDM method we still need to set bunch of hyperparameters like scaling factor c, momentum coefficient v. Therefore FSGDM is not very different from the tradition momentum method in terms of hyperparameter tuning.
  2. The optimal value of scaling factor c and momentum coefficient v is determined for different datasets and network architectures (as shown in fig 4) and is used for the results shown in the experiment section. While some standard coefficients are used for standard SGDM and EMA SGDM. This shows that the results might be slightly biased.
* The frequency domain analysis in this paper remains somewhat limited in its applicability to broader optimization contexts. The framework primarily addresses momentum-based methods in the context of SGD-like algorithms, but it does not extend to other popular optimization strategies like Adam or RMSprop, which also incorporate adaptive mechanisms and momentum-like components. Therefore I suggest extending the frequency domain analysis for other popular optimization methods like Adam and RMSprop would make this paper much more significant.

**Questions:**

Suggestions:
* It’s unclear how robust the optimizer is to hyperparameter tuning across different datasets and tasks. Without this information, practitioners might face challenges in adopting FSGDM in real-world applications. Therefore I suggest to include a dedicated analysis or appendix section detailing the sensitivity of FSGDM’s performance to variations in these hyperparameters. It would be useful to identify hyperparameter settings that generalize well across tasks, if any.

Questions:
* Since Adam and RMSprop uses both gradient and curvature of the error surface
  ** how might the frequency domain analysis framework be extended for Adam or RMSprop?
  ** What challenges do you anticipate in applying this framework to those methods?
  ** Do you believe similar insights about gradient frequency components would apply, or would different principles emerge?

---

> ### Author Response · Authors · 2024-11-20
>
> Thank you for the thoughtful review, valuable feedback, and acknowledgement of the significance and originality of our work. We provide point-by-point clarifications and answer the questions below.
>
> **Weaknesses**:
>
> **W1.1: The setting of hyperparameters is still required for the proposed FSGDM**.
>
> We agree that using FSGDM requires setting the scaling factor c and the momentum coefficient v. However, it is important to note that tuning c and v involves finding configurations that yield favourable dynamic magnitude responses for FSGDM. One of our main contributions is demonstrating, through our proposed frequency domain analysis framework, that hyperparameters yielding better test performance exhibit similar dynamic magnitude responses, as shown in Figure 13. Moreover, this pattern generalizes well across various tasks and generally outperforms existing methods. Therefore, we believe our work will still provide valuable insights and inspiration to the community.
>
> **W1.2: The results of standard SGDM and EMA SGDM might be slightly biased without tuning**.
>
> We address this issue from two perspectives:
>
> 1. The hyperparameters of FSGDM, i.e. $c = 0.033$, $v = 1$, were conveniently selected from the black points within the optimal zone identified in Section 4.2 based on our proposed framework, and these hyperparameters were consistently used across all the experiments we conducted. As shown in Figure 14. (b) of Appendix B.8, there are other hyperparameters within the optimal zone that can lead to better results. All these hyperparameters are within the optimal zone, and they have similar dynamic magnitude responses.
>
> 2. We used the standard momentum coefficient, i.e. $u_t=0.9$, for both EMA-SGDM and Standard-SGDM for the following four reasons. (1). In Figure 11 in Appendix B.6, we showed that the test performances for both these two optimizers decline when $u_t > 0.9$; (2) Tables 1 and 6 illustrated that there is no consistent trend for the test performance using EMA-SGDM when $u_t = 0.3, 0.6, 0.9$; (3) Tables 2 and 7 showed that the test performance gradually increases using Standard-SGDM when $u_t$ increases among $0.3$, $0.6$, $0.9$; (4) the differences in performance due to varying hyperparameters are much smaller than the performance gap observed when compared to FSGDM. Therefore, in our experiments, we selected the commonly used $u_t = 0.9$ for both EMA-SGDM and Standard-SGDM.
>
> **W2: Frequency domain analysis for Adam and RMSprop**.
>
> Since this issue is similar to the question in **Q2**, we kindly refer the reviewer to our response to **Question 2**.
>
> **Questions**:
>
> **Q1: The robustness of FSGDM is unclear**.
>
> Thanks for this excellent question and we agree this is of great importance. In Figure 4, we conducted experiments on three different tasks using three different models. By grid-searching the test accuracy under various hyperparameter selections and plotting the heat map, we discover a similar optimal zone across different datasets. The hyperparameters within the optimal zone show high similarity in terms of test accuracy due to their similar dynamic magnitude responses, as shown in Figure 13. While we recognize the importance of more extensive experiments, we believe that the existing explorations regarding the optimal zone and FSGDM's sensitivity to variations in hyperparameters still provide valuable insights and a strong foundation for future works.
>
> **Q2: Application to Adam and RMSprop**.
>
> Thank you for raising this important question about extending our proposed frequency-domain analysis framework to other optimizers like Adam. However, this extension is not straightforward. While the first-moment estimate of Adam is in the form of EMA and thus acts as a low-pass filter, the second-moment estimate presents additional obstacles for frequency domain analysis in the following ways:
>
> 1. The second-moment estimates of Adam and RMSprop involve the squared gradient term $g_t^2$, resulting in nonlinearity that complicates the direct application of the Z-transform.
>
> 2. Adam introduces both the first- and second-moment estimates ($m_t$ and $v_t$) and adopts $\hat{m}_t/(\sqrt{\hat{v}_t}+\epsilon)$ as the update step. This intricate interaction between $m_t$ and $v_t$ also makes the analysis more challenging.
>
> At this stage, we believe that our argument regarding the three insights discussed in Section 3.3 is also applicable to other optimizers. However, how the different frequency gradient components in the model parameter updates are processed by the Adam optimizer remains unclear. We anticipate that resolving these issues will provide deeper insights.

---

> > ### Author Response · Authors · 2024-11-25
> > **Additional Experiments for Q1**
> >
> > We explored the optimal hyperparameter selection of FSGDM for the IWSLT14 translation task by training LSTM-W and Transformer-tiny. We swept the hyperparameters of FSGDM, i.e. $c$ and $v$, within the range of $(0,1)$ and $[0.5,3]$. The detailed experimental settings follow Appendix C. We have added these additional experiments in Appendix B.7.
> >
> > The results in Appendix B.7 indicate that similar optimal zones can be observed on the NLP task. When the momentum coefficient $v$ is fixed, the BLEU score shows an initial increase followed by a decline as the scaling factor $c$ increases, which is highly consistent with the results in Section 4.2. In addition, we find that the empirical insights discussed in Section 3.3 are also applicable to various deep learning models beyond CNNs, as well as NLP tasks.
> >
> > In conclusion, Figures 4 and 12 explore a wide range of hyperparameter selections across different datasets, tasks and deep learning models to demonstrate how sensitive FSGDM’s performance is to variations in these hyperparameters.

---

> > > ### Comment · Reviewer_RGqa · 2024-11-26
> > > **Response to Authors**
> > >
> > > Thanks for performing additional experiments for W1.2: This shows that even with different setting of momentum coefficient FSGDM outperforms standard SGDM and EMA-SGDM on CIFAR-100 ResNet-50.
> > >
> > > Thanks for clarifying other questions.

---

> > > > ### Author Response · Authors · 2024-11-26
> > > > **Response to Reviewer RGqa**
> > > >
> > > > Thank you for getting back to us! We are glad that the additional experiments were helpful. We hope that our answers have addressed all of your concerns. If you have any further questions or if there are additional points you'd like us to clarify, please do not hesitate to let us know. We would be happy to provide more details. If you are satisfied with our responses, we hope you will consider raising your score.

---

> ### Author Response · Authors · 2024-11-23
> **Looking Forward to Further Feedback**
>
> Dear Reviewer RGqa,
>
> Thank you once again for your valuable comments. Please let us know if our responses have addressed your concerns. As the deadline for the discussion phase approaches, we welcome any additional feedback and are happy to clarify any remaining issues.
>
> Best,
>
> Authors

---

> ### Author Response · Authors · 2024-11-24
> **Additional Experiment for W1.2 (1/n)**
>
> We trained ResNet-50 on the CIFAR-100 dataset using EMA-SGDM and Standard-SGDM and swept momentum coefficient $u_t$ over 9 values ranging from 0.1 to 0.9. For each experiment, we reported the mean and standard error (noted as subscripts) of test accuracy for 3 runs with seeds 0-2. The detailed experimental settings are outlined in Appendix C.
>
> The table below presents the test accuracy results. Our findings confirm two key points: (1). For EMA-SGDM, there is no consistent trend for the test performance when $u_t$ increases from 0.1 to 0.9; (2). For Standard-SGDM, the test performance demonstrates a generally increasing trend as $u_t$ rises from 0.1 to 0.9. Based on these observations, we chose the commonly used value of $u_t = 0.9$ for both EMA-SGDM and Standard-SGDM in our experiments.
>
> | $u_t$ | 0.1 | 0.2 | 0.3 | 0.4 | 0.5 | 0.6 | 0.7 | 0.8 | 0.9 |
> | ------ | ----- | ----- | ----- | ----- | ----- | ----- | ----- | ----- | ----- |
> | EMA-SGDM | $76.97_{0.04}$ | $76.77_{0.16}$ | $76.98_{0.09}$ | $76.79_{0.21}$ | $76.96_{0.07}$ | $76.82_{0.18}$ | $76.91_{0.23}$ | $76.87_{0.10}$ | $76.84_{0.06}$ |
> | Standard-SGDM |  $77.68_{0.14}$ | $77.90_{0.17}$ | $78.01_{0.04}$ | $78.51_{0.19}$ | $79.05_{0.45}$ | $79.51_{0.15}$ | $79.58_{0.08}$ | $79.42_{0.11}$ | $79.71_{0.25}$ |

---

> > ### Author Response · Authors · 2024-11-27
> > **Additional Experiment for W1.2 (2/n)**
> >
> > In addition, we trained LSTM-W and Transformer-tiny on the IWSLT14 dataset using EMA-SGDM and Standard-SGDM and swept momentum coefficient $u_t$ over 9 values ranging from 0.1 to 0.9. For each experiment, we reported the BLEU score for one run with seed 0. The detailed experimental settings are outlined in Appendix C.
> >
> > The tables below present experimental results. Our findings confirm three key points: (1). For EMA-SGDM, there is no consistent trend for the BLEU score when $u_t$ increases from 0.1 to 0.9; (2). For Standard-SGDM, the BLEU score demonstrates a generally increasing trend as $u_t$ rises from 0.1 to 0.9; (3) Standard-SGDM consistently outperforms EMA-SGDM by a large margin. Based on these observations, we chose the popular value of 0.9 for both EMA-SGDM and Standard-SGDM in our NLP experiments.
> >
> > **LSTM-W**:
> >
> > | $u_t$ | 0.1 | 0.2 | 0.3 | 0.4 | 0.5 | 0.6 | 0.7 | 0.8 | 0.9 |
> > | ------ | ----- | ----- | ----- | ----- | ----- | ----- | ----- | ----- | ----- |
> > | EMA-SGDM | 1.43 | 1.45 | 1.39 | 1.40 | 1.44 | 1.46 | 1.36 | 1.38 | 1.31 |
> > | Standard-SGDM | 1.54 | 1.77 | 2.24 | 2.50 | 3.08 | 4.30 | 7.66 | 20.64 | 24.73 |
> >
> > **Transformer-tiny**:
> >
> > | $u_t$ | 0.1 | 0.2 | 0.3 | 0.4 | 0.5 | 0.6 | 0.7 | 0.8 | 0.9 |
> > | ------ | ----- | ----- | ----- | ----- | ----- | ----- | ----- | ----- | ----- |
> > | EMA-SGDM | 5.39 | 5.38 | 5.31 | 5.27 | 5.27 | 5.29 | 5.31 | 5.34 | 5.20 |
> > | Standard-SGDM | 5.85 | 6.38 | 6.94 | 7.54 | 8.22 | 9.28 | 10.99 | 13.71 | 18.20 |
> >
> > With these additional experiments in CV and NLP tasks, we show that the momentum coefficient selection for EMA-SGDM and Standard-SGDM is reasonable. The conclusion that FSGDM can consistently outperform EMA- and Standard-SGDM is not affected.

---

### Official Review · Reviewer_Whrt · 2024-11-02

**Soundness:** 3
**Presentation:** 2
**Contribution:** 3
**Rating:** 6
**Confidence:** 1

**Summary:**

This paper introduces a frequency-domain analysis framework for understanding momentum-based optimizers, viewing them as time-variant filters that adaptively shape gradient characteristics through momentum coefficients. Based on these insights, the authors propose FSGDM, a novel optimizer that dynamically adjusts momentum to enhance generalization, outperforming conventional momentum-based methods in experiments.

**Strengths:**

1. The paper is clear and straightforward, making it easy to follow. The core idea is both simple and effective, showcasing an elegant solution that achieves strong results without unnecessary complexity.

**Weaknesses:**

1. In traditional signal processing, many derivations provided in the paper are well-established and widely recognized.
2. Quantitative comparisons within this field remain relatively limited, often lacking comprehensive metrics or benchmarks to evaluate performance across different methods.

**Questions:**

Why the Z-transform is employed in the learning settings, while many other transforms can also be deployed?

---

> ### Author Response · Authors · 2024-11-16
>
> Thank you sincerely for these valuable comments and acknowledgement of our work. To ensure clarity and address your concerns comprehensively, we respond to each comment individually:
>
> **Weaknesses**:
>
> **W1**: While we acknowledge that the derivations in our paper are based on well-established signal processing techniques, we believe our work still makes a significant contribution to the field of machine learning. In our frequency domain analysis framework, we adapt effectively the signal processing theories to the analysis of the momentum method. To the best of our knowledge, it is the first time that a frequency domain analysis framework has been proposed to offer a deeper, more intuitive understanding of the momentum mechanism in training deep neural networks. This new perspective not only enhances the analysis of training processes but also offers valuable insights that could inspire further research in this area. Therefore, we believe our work offers a significant contribution to the machine learning community.
>
> **W2**:  We agree that comprehensive metrics and benchmarks are crucial for thorough evaluation. In our paper, we followed common practices in the field as demonstrated in previous works [1-4], when selecting metrics and conducting comparisons. Additionally, in Appendix B, we present further experiments and a broader range of comparisons, which offer additional validation of our frequency-domain framework. While we recognize the importance of a broader evaluation framework, we believe that our analysis still provides valuable insights and a strong foundation for future research to build upon.
>
> **Questions**:
>
> Aside from the Z-transform, there are many transform techniques in signal processing, e.g., Fourier transform, Laplace transform, and Mellin transform, to name a few, but most of them are not appropriate for analyzing the momentum mechanism for the following reasons.
>
> 1. The momentum updates in Eqn.(1) and (2) are in the discrete-time domain. Therefore, those continuous-time transforms, such as Laplace transform, Mellin transform, etc., are not directly applicable to discrete-time gradient and momentum updates.
>
> 2. Some discrete-time transforms, such as the Discrete Cosine Transform (DCT) and the Hilbert Transform, can convert signals from the time domain to the frequency domain. The DCT is primarily used in image processing and audio compression, while the Hilbert Transform is mostly used for analyzing instantaneous frequency and envelope extraction. To conclude, these transform methods are not the ideal choices for analyzing recursive momentum systems due to their different application areas.
>
> 3. The Discrete-Time Fourier Transform (DTFT) and the Z-transform are commonly used for analyzing discrete-time recursive systems. The Z-transform is defined for all values of $z$ in the complex plane, whereas the DTFT only concerns values of $z$ that lie on the unit circle, i.e., when $z=e^{j\omega}$. When the Z-transform is evaluated on the unit circle in the complex plane, it degenerates into DTFT, which provides the frequency spectrum of the discrete-time signal. Thus, the DTFT is a special case of the Z-transform. Consequently, we adopted Z-transform as the general solution.
>
> In conclusion, the Z-transform is widely used for recursive discrete-time systems including momentum systems. It is also a general version of DTFT. Therefore, we employed Z-transform in our analysis.
>
>
> Please do not hesitate to contact us if you have any further questions regarding our work. We are more than willing to provide additional details and discuss our approach further.
>
> **References**:
>
> [1]. Defazio, Aaron, Xingyu Alice Yang, Harsh Mehta, Konstantin Mishchenko, Ahmed Khaled, and Ashok Cutkosky. "The road less scheduled." In The Thirty-eighth Annual Conference on Neural Information Processing Systems, 2024.
>
> [2]. Xie, Xingyu, Pan Zhou, Huan Li, Zhouchen Lin, and Shuicheng Yan. "Adan: Adaptive nesterov momentum algorithm for faster optimizing deep models." IEEE Transactions on Pattern Analysis and Machine Intelligence, 2024.
>
> [3]. Liu, Yanli, Yuan Gao, and Wotao Yin. "An improved analysis of stochastic gradient descent with momentum." In The Thirty-fourth Annual Conference on Neural Information Processing Systems, 2020.
>
> [4]. Wang, Runzhe, Sadhika Malladi, Tianhao Wang, Kaifeng Lyu, and Zhiyuan Li. "The Marginal Value of Momentum for Small Learning Rate SGD." In The Twelfth International Conference on Learning Representations, 2024.

---

> ### Author Response · Authors · 2024-11-22
> **Looking Forward to Further Feedback**
>
> Dear Reviewer Whrt,
>
> Thank you once again for your valuable comments. Please let us know if our responses have addressed your concerns. As the deadline for the discussion phase approaches, we welcome any additional feedback and are happy to clarify any remaining issues.
>
> Best,
>
> Authors

---

> > ### Comment · Reviewer_Whrt · 2024-11-26
> >
> > I am not an expert in this area, I tend to leave the decision to other reviewers and AC.

---

### Official Review · Reviewer_AKLA · 2024-11-04

**Soundness:** 3
**Presentation:** 4
**Contribution:** 4
**Rating:** 8
**Confidence:** 4

**Summary:**

The paper proposes a signal processing framework for the analysis of momentum in stochastic gradient descent (SGD). Specifically, the authors consider two variants of momentum SGD: coupled (where the previous and the current step proportions add up to one) and decoupled (where the current step has the weight of one, and the previous step can have the momentum term tweaked independently). Furthermore, the authors use the signal processing viewpoint to gain insight on how the momentum coefficients should evolve over the course of neural network training. Finally, based on the insights regarding the training behavior under various momentum settings, the authors propose a new frequency-based variant of momentum SGD, and show on a wide variety of problems that the proposed variant outperforms the conventional momentum SGD (both coupled and decoupled).

**Strengths:**

**Originality:** The paper brings the signal processing perspective to the understanding of SGD, which is invaluable. Seeing that a lot of the progress in ML is purely empirical, the systematic and thorough approach the paper takes is both refreshing and useful. Viewing momentum term contributions as corresponding to low/high-pass filters opens the door to apply various insights from signal processing directly to neural network training. Thinking of momentum as attenuating and amplifying various gradient frequencies also provides a useful intuition, and maps well to existing state-of-the-art optimizers such as Adam.

**Observations:** The authors note that decoupled rather than coupled approach to momentum is more beneficial. Further, the authors observe the following set-up as beneficial to generalization performance: retaining the original gradient early in training; focusing on high-frequency gradients early in training, and suppressing them closer to the end of training; amplifying low-frequency gradients at the end of training. Perhaps the insights gained can lead to further studies into generalization behavior of neural networks.

**Quality and Clarity:** The paper is very well-written, with only a few minor typos. The methodology is clear. A lot of the results are pushed into appendices, which indicates that the authors have done an excessive amount of work towards the paper. I feel bad that such extensive experimentation is becoming a de facto standard for top-tier conferences. Nevertheless, there is evidence that the authors went above and beyond the expectation.

**Significance:** The paper asks fundamental questions, and provides useful answers to these questions. It offers a significant new perspective that can aid the design of new, more efficient optimizers.

**Weaknesses:**

An important work on how momentum works was published in 2017 in the Distill web-based journal: https://distill.pub/2017/momentum/
I believe that this paper needs to feature in the related literature. Also, in case the authors are not aware of it, I believe they will enjoy reading it!

As mentioned above, the experiments conducted are truly comprehensive and span a wide range of problems and architectures. However, what is missing from the experiments is the presence of the (arguably) state of the art approach to NN training, i.e., the Adam optimizer. For the reinforcement learning experiments, the authors explicitly state that they have replaced Adam with their method as well as coupled and decoupled versions of momentum SGD. However, based on the insights, it seems that Adam succeeds by doing exactly what the paper prescribes as necessary: amplify low-frequency gradients while attenuating high-frequency gradients. A discussion of Adam in the context of the proposed paradigm, even on a superficial level, would benefit the discussion, in my opinion.

**Questions:**

A few mistakes that I picked up need to be fixed:
* Page 2, line 95: SGD-base -> SGD-based
* Page 6, line 321: “test results of orthodox momentum…” - do you mean unorthodox?
* Page 7, line 351: “A Large” - “large” should not be capitalized

Legend on the figures is in a very small font, which makes it rather hard to read.

Tables 1 and 2 refer to various "Experiments" (1, 2, 3, and 4). While I realize that the experiments are defined in the main body of the text, I think the tables can be made more easily interpretable on their own by rather using a descriptive name (high-pass VS low-pass gain, etc.) per row.

---

> ### Author Response · Authors · 2024-11-20
>
> We appreciate the reviewer's acknowledgement of our work and the valuable feedback. To address the concerns raised about the paper, we have provided the following responses.
>
> **Weaknesses**:
>
> **W1: Related work**.
>
> Thank you for bringing the article on momentum (https://distill.pub/2017/momentum/) to our attention. We agree that it is a significant work providing valuable background and insights into the momentum mechanism in optimization. We have now included a citation to this paper in our manuscript to further enrich the context and enhance readers' understanding.
>
> **W2: Relation with Adam**.
>
> In this paper, we primarily focus on the first-moment structure, which is why we have not included experiments with Adam, as it introduces a second-moment structure.
>
> While the first-moment structure in Adam attenuates high-frequency gradients, it does not amplify low-frequency gradients but rather preserves them to some extent. This property endows the first-moment structure of Adam with low-pass filtering characteristics, similar to EMA-SGDM. Moreover, due to the complicated second-moment structure and its interaction with the first-moment structure in Adam, simply applying our current analysis framework to Adam is challenging as mentioned in our answer to **Question 2** raised by **Reviewer RGqa**. For the page limitation and the complexity of extending our framework to accommodate the second-moment structure, we decide to leave it for future work. Accordingly, we have expanded our paper to include additional content on this topic. We hope this addition clarifies your concerns, and we welcome any further questions or feedback you may have.
>
> **Questions**:
>
> We appreciate your attention to these details and have modified our paper accordingly. We believe these revisions improve the clarity and quality of our manuscript.

---

> > ### Comment · Reviewer_AKLA · 2024-11-20
> > **Satisfied with the changes**
> >
> > Thank you for updating the paper, I have gone through it again and am happy with the minor corrections. I have spotted another typo in the appendices: Appendix C, line 1057 - “Date Augmentation” must surely read as “Data Augmentation”.

---

> > > ### Author Response · Authors · 2024-11-20
> > >
> > > Thank you for revisiting our paper and for pointing out the typo in Appendix C. We appreciate your careful reading and attention to detail. We have corrected the typo in the revised version as you suggested.

---

### Author Response · Authors · 2024-11-20
**General Response**

We would like to express our sincere gratitude to all the reviewers for their insightful feedback and constructive comments. We especially appreciate the reviewers' acknowledgement of the significance and novelty of our work.

We have carefully considered and responded to all the comments and made the necessary revisions accordingly. Here, we summarize the changes made to the paper:

1. We have fixed the typos in the paper, thanks to the feedback from Reviewer AKLA.

2. We have enlarged the font size of the figure legends, updated the labels in Tables 1, 2, 6, and 7 for clearer understanding, and added the related work to enhance the completeness of our paper as suggested by Reviewer AKLA.

3. We have updated the results in Table 4 after discovering that the parameters for EMA-SGDM were inadvertently set the same as those for Standard-SGDM while cleaning up our code. This correction indicates that EMA-SGDM's performance is lower than that of Standard-SGDM, which is consistent with our other experiments. Importantly, this error does **NOT** affect any analyses or conclusions in our paper. All other experiments have been checked for correctness and we will open-source the code if the paper is accepted. We are sincerely sorry for any inconvenience caused.

4. In response to Reviewer AKLA and Reviewer RGpa, we have added a discussion in Appendix D addressing the relationship between our framework and adaptive optimizers like Adam, as well as the challenges associated with their analysis.

5. In response to Reviewer RGpa, we have included additional experiments in Appendix B.7 that explore the optimal parameter selection of FSGDM for the NLP task by training LSTM-W and Transformer-tiny.

6. We have corrected grammatical errors and polished the writing in the paper.

---

### Author Response · Authors · 2024-11-26
**Following up on Reviewers' Concerns**

Dear Reviewers and Area Chair,

Thank you once again for your time and effort in reviewing our paper and for going through our responses to your questions. As the PDF revision deadline approaches, we would like to know if we have adequately addressed your concerns or if you have any further questions. We are more than happy to respond and make any necessary revisions to our paper.

Best regards,

Authors of Paper 5958

---

### Meta-Review · Area_Chair_fXvm · 2024-12-20

**Metareview:**

This paper studies the widely used momentum methods for neural network training. It tries to give more insights about different forms of momentum methods and gives a new explanation of momentum methods from the frequency domain. It also introduces a practical  approach to adjust the momentum parameter. The paper has also conducted extensive empirical studies. All reviewers agree the paper is novel and bring new insights to momentum method. Hence, AC recommends an acceptance.

**Additional Comments On Reviewer Discussion:**

The reviewers interact with authors during the discussion phase and acknowledge that their concerns have been addressed.

---

### Decision · Program_Chairs · 2025-01-22

Accept (Poster)